# Broad CD8+ T cell cross-recognition of distinct influenza A strains in humans

Emma J. Grant[1,2], Tracy M. Josephs[2], Liyen Loh[1], E. Bridie Clemens[1], Sneha Sant[1], Mandvi Bharadwaj[1], Weisan Chen[3], Jamie Rossjohn ⓘ [2,4,5], Stephanie Gras[2,5] & Katherine Kedzierska[1]

Newly-emerged and vaccine-mismatched influenza A viruses (IAVs) result in a rapid global spread of the virus due to minimal antibody-mediated immunity. In that case, established CD8+ T-cells can reduce disease severity. However, as mutations occur sporadically within immunogenic IAV-derived T-cell peptides, understanding of T-cell receptor (TCRαβ) cross-reactivity towards IAV variants is needed for a vaccine design. Here, we investigate TCRαβ cross-strain recognition across IAV variants within two immunodominant human IAV-specific CD8+ T-cell epitopes, HLA-B*37:01-restricted $NP_{338-346}$ (B37-$NP_{338}$) and HLA-A*01:01-restricted $NP_{44-52}$ (A1-$NP_{44}$). We find high abundance of cross-reactive TCRαβ clonotypes recognizing distinct IAV variants. Structures of the wild-type and variant peptides revealed preserved conformation of the bound peptides. Structures of a cross-reactive TCR-HLA-B37-$NP_{338}$ complex suggest that the conserved conformation of the variants underpins TCR cross-reactivity. Overall, cross-reactive CD8+ T-cell responses, underpinned by conserved epitope structure, facilitates recognition of distinct IAV variants, thus CD8+ T-cell-targeted vaccines could provide protection across different IAV strains.

[1] Department of Microbiology and Immunology, The University of Melbourne, at the Peter Doherty Institute for Infection and Immunity, Melbourne, VIC 3010, Australia. [2] Infection and Immunity Program and Department of Biochemistry and Molecular Biology, Biomedicine Discovery Institute, Monash University, Clayton, VIC 3800, Australia. [3] Department of Biochemistry and Genetics, La Trobe Institute of Molecular Science, La Trobe University, Bundoora 3084 VIC, Australia. [4] Institute of Infection and Immunity, School of Medicine, Cardiff University, Cardiff CF14 4XN, UK. [5] ARC Centre of Excellence in Advanced Molecular Imaging, Monash University, Clayton, VIC 3800, Australia. These authors contributed equally: Stephanie Gras, Katherine Kedzierska. Correspondence and requests for materials should be addressed to S.G. (email: stephanie.gras@monash.edu) or to K.K. (email: kkedz@unimelb.edu.au)

nfluenza A viruses (IAVs) rapidly evolve and cause significant morbidity and mortality (reviewed in refs. [1,2]). Annual epidemics are responsible for >500,000 deaths worldwide[3], while pandemics can cause >50 million deaths (reviewed in ref. [4]). Although vaccines are available, they primarily induce neutralizing antibodies directed towards the rapidly mutating surface glycoproteins, rather than cross-reactive CD8+ T cell immunity[1,5], mandating that these vaccines are updated and administered annually (reviewed in ref. [6]). Furthermore, these vaccines are fallible when the circulating strains do not match the predicted vaccine strains[7] or in a scenario when a novel viral subtype enters the population. Thus there is an urgent need to understand correlates of T cell protection towards IAV to provide effective influenza vaccine design.

In the absence of neutralizing antibodies, strain cross-reactive CD8+ T cells can protect against IAVs. Murine studies show that CD8+ T cells correlate with decreased morbidity and mortality following IAV infection[8–12] and can provide protection during infection with heterosubtypic IAV strains[11,13–15]. Human studies are consistent with murine data. Namely, published evidence shows that prominence of influenza-specific CD8+ T cells correlates with lower viral titers[16] and decreased disease severity[17–19] during IAV infection. Furthermore, CD8+ T cells primed with seasonal circulating IAV strains can cross-react with pandemic H1N1 (pH1N1) or variant seasonal peptides[20–22] or virulent H7N9 and H5N1 avian IAV-derived peptides[23–26]. Together, these data suggest that an IAV-specific CD8+ T cell-mediated vaccine can provide broad cross-reactive immunity across distinct influenza A strains and subtypes for both conserved and variable CD8+ T cell epitopes.

It is well established that CD8+ T cells with diverse T cell receptor (TCR) repertoires are greatly beneficial for disease outcome, contributing to reduced disease severity[27], enhanced CD8+ T cell function[28], cross-reactivity across different peptide variants[29,30], and preventing viral escape[31,32]. Importantly, although CD8+ TCRs are typically highly specific for their cognate peptide, they can also recognize a broad range of peptide variants, thus allowing CD8+ T cells to have a powerful capacity to recognize not only their cognate peptide but also a range of viral mutants[11,30,33–36]. In case of highly mutating influenza viruses, such cross-reactive CD8+ T cells are highly desirable as they elicit immune responses towards multiple viral strains and hence provide cross-strain protection.

The precise mechanisms underlying cross-recognition by influenza-specific CD8+ TCRs in humans are unclear. To date, TCRαβ repertoires have only been dissected for two immunodominant influenza-specific human epitopes, HLA-A*02:01-restricted $M1_{58}$[30] and HLA-B*35:01/*35:03/*07:02-restricted $NP_{418}$[30], providing ~50% of the cumulative population coverage. Thus it is important to understand cross-reactivity and diversity of CD8+ T cell TCRαβ repertoires directed against other prominent IAV-specific epitopes, if we are to rationally design a broadly protective CD8+ T cell-mediated influenza vaccine.

Here we use an ex vivo multiplex reverse transcription polymerase chain reaction (RT-PCR) approach[30,37,38] to analyze paired TCRαβ repertoires for two additional prominent human CD8+ T cell epitopes, HLA-B*37:01-restricted $NP_{338–346}$-FEDLRVLSF ($NP_{338}$)[39] and HLA-A*01:01-restricted $NP_{44–52}$ CTELKLSDY ($NP_{44}$)[23,40], restricted by alleles that are frequent in the human population (~19% of the cumulative coverage). We identify cross-reactive TCRαβ clonotypes capable of recognizing the wild-type (WT) peptide and peptide variants. This is most prominent in HLA-B*37:01-expressing donors, where distinct and cross-reactive $NP_{338}$-specific TCRαβ clonotypes bound each of the $NP_{338}$-WT, $NP_{338}$-L7S, and $NP_{338}$-V6L variants (93–100% of distinct IAV strains), highlighting their potential to provide protection against distinct influenza strains and subtypes. Our structural analysis reveals that the variants adopt a similar conformation than the WT epitope for both HLA-A*01:01 (HLA-A1)

and HLA-B*37:01 (HLA-B37) molecules, providing a molecular basis for CD8+ TCRαβ cross-reactivity. Structural analysis indicates that molecular similarity may underpin how an HLA-B37-restricted cross-reactive TCRαβ, clone EM2, can recognize the variants. Thus our data suggest that structural resemblance underpins cross-reactivity of HLA-B37+$NP_{338}$+CD8+ and HLA-A1+$NP_{44}$+CD8+ T cells, despite their diverse TCR repertoires between individuals towards those two epitopes.

## Results

**Only HLA-B37+ donors elicit a $NP_{388}$+CD8+ T cell response.** Our previous work identified $NP_{338}$ as an immunodominant CD8+ T cell epitope in individuals expressing HLA-B37[39]. However, $NP_{338}$ was previously reported to be restricted by HLA-B*44:03[41]. Thus we first investigated whether the $NP_{338}$ peptide can be presented by other HLA allomorphs using the online SYFPETHI peptide-binding prediction tool[42]. A high prediction binding score for the $NP_{338}$ peptide was obtained for HLA-B37, HLA-B*44:02, and HLA-B*18:01 (HLA-B18) (Table 1). The SYFPETHI website contains only the HLA-B*44:02 in its database but not the closely related allomorphs, such as HLA-B*44:03 and HLA-B*44:05[43]. The high prediction binding score for the $NP_{338}$ peptide obtained for HLA-B*44:02 is likely to be shared by HLA-B*44:03 and HLA-B*44:05 (HLA-B44), given their known overlapping peptide repertoire[44]. Both HLA-B44[44–47] and HLA-B18[46,48] display a preference for P2-E and PΩ-F/Y in their bound peptides, and both residues are present in the IAV-derived $NP_{338}$ peptide.

We first refolded the $NP_{338}$ peptide with HLA-B37, HLA-B18, and HLA-B44 molecules and assessed the stability of each peptide–HLA (pHLA) complex. We used HLA-B*44:05 molecule as the refold yield was higher for this allomorph than for the HLA-B*44:02/03. The $NP_{338}$ peptide could bind each of the three HLA molecules (HLA-B37, -B18, and -B*44:05); however, the stability of HLA-B37-$NP_{338}$ was superior by 6 °C to HLA-B18-$NP_{338}$ and by 14 °C to HLA-B*44:05-$NP_{338}$ (Supplementary Table 1).

We then solved the binary structures of the $NP_{338}$ peptide presented by HLA-B37, HLA-B18, and HLA-B44 (Supplementary Table 2, Fig. 1). The $NP_{338}$ peptide adopted an extended conformation in the cleft of HLA-B37, with the P2-Glu and P9-Phe acting as anchor residues in the B- and F-pocket of HLA-B37, respectively[49]. In addition to the primary anchor residues, the P5-Arg acted as a secondary anchor residue, fully buried into the F-pocket, and formed a salt bridge with Asp77 from the α1-helix in the HLA-B37 molecule (Fig. 1a). The remaining six residues were

**Table 1 HLA alleles predicted to bind the WT $NP_{338–346}$ FEDLRVLSF peptide**

| Prediction score range | HLA allele | Prediction score | HLA superfamily |
|---|---|---|---|
| >20 | **B*37** | **33** | **B44** |
| | **B*44:02** | **25** | **B44** |
| | **B*18** | **22** | **B44** |
| 15–20 | B*08 | 17 | B8 |
| | B*40:01 | 17 | B44 |
| | B*45:01 | 17 | B44 |
| | B*41:01 | 16 | ND |
| | B*14:02 | 15 | B27 |
| | B*53:01 | 15 | B7 |
| | B*38:01 | 15 | B27 |
| | B*13 | 3 | ND |
| | A*68:01 | 2 | A3 |

The http://www.syfpeithi.de/bin/MHCServer.dll/Info.htm prediction tool was used to assign a prediction score that the WT $NP_{338}$ peptide would be bound by the particular HLA allele. Prediction scores were grouped into a predicted score range for clarity. ND refers to HLA alleles, which have not been assigned a HLA superfamily. Alleles in bold were selected for further analysis

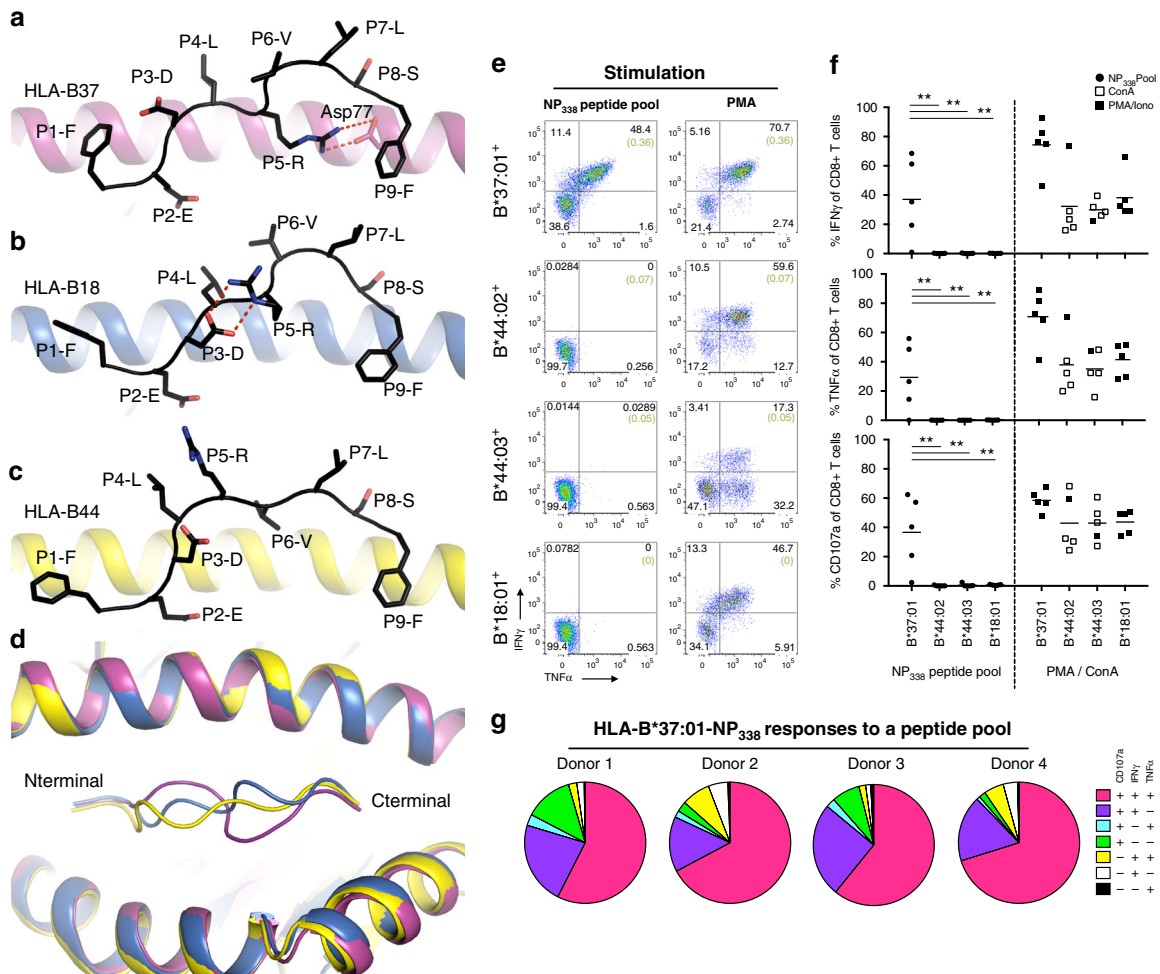

**Fig. 1** NP$_{338}$ elicits robust CD8$^+$ T cell responses only in the context of HLA-B*37:01. **a–d** Crystal structures of **a** HLA-B*37:01 (purple), **b** HLA-B*18:01 (blue), and **c** HLA-B*44:05 (yellow) in complex with NP$_{338}$ WT (represented in black stick). **d** is showing an overlay of the three peptide–HLA complexes in cartoon representation, with NP$_{338}$ colored accordingly to the bound HLA, namely, purple (HLA-B*37:01), blue (HLA-B*18:01), and yellow (HLA-B*44:05). **e–g** PBMCs from HLA-B*37:01$^+$ donors ($n = 5$), HLA-B*44:02$^+$ donors ($n = 5$), HLA-B*44:03$^+$ donors ($n = 5$), or HLA-B*18:01$^+$ donors ($n = 5$) were stimulated with a pool of NP$_{338}$ peptides (10 µM each peptide, 30 µM total), and specificity was assessed in an ICS assay against 3 µM NP$_{338}$ pool (1 µM each peptide, HLA-B*44:02$^+$ donors) directly into the well or 30 µM (10 µM each peptide, all other donors) NP$_{338}$-pooled pulsed APC lines (HLA-B*37:01, HLA-B*44:03, and HLA-B*18:01). CD8$^+$ T cells are gated on lymphocytes, singlets, live$^+$ (when stained with the second antibody cocktail only) CD3$^{mid-high}$, CD8$^+$ T cells; as per Supplementary Fig. 2a. **e** Representative dot plots of IFNγ$^+$TNFα$^+$ production by pooled CD8$^+$ T cell lines derived from HLA-B*37:01$^+$, HLA-B*44:02$^+$, HLA-B*44:03$^+$, or HLA-B*18:01$^+$ donors, towards a pool of NP$_{338}$ peptides. No peptide control represented in brackets. **f** Summary of IFNγ, TNFα, and CD107a production, minus no peptide controls, against the pool of NP$_{338}$ peptides (closed circle) or a positive control (ConA open square or PMA closed square) across multiple donors. Statistical analysis using a Dunnett's two-way ANOVA in which *$p \leq 0.05$, **$p \leq 0.01$, and ***$p \leq 0.001$. **g** Summary of the polyfunctional profiles, minus background, from the four HLA-B*37:01$^+$ donors who responded towards the pool of NP$_{338}$ peptides

all solvent exposed and represented potential contacts for the TCR. Comparison of the HLA-B37-NP$_{338}$ with HLA-B18-NP$_{338}$ and HLA-B44-NP$_{338}$ complexes show an overall similar HLA-binding cleft (Fig. 1d), with a root mean square deviation (r.m.s. d.) of 0.2 and 0.4 Å, respectively. Similar to the HLA-B37-NP$_{338}$ structure, the NP$_{338}$ peptide was anchored to HLA-B18 and HLA-B44 by P2-Glu and P9-Phe. However, the NP$_{338}$ peptide adopted a strikingly distinct conformation when bound to HLA-B37 as compared with HLA-B18 (r.m.s.d. of the peptide 1.7 Å) and HLA-B44 (r.m.s.d. of 1.8 Å for the peptide). In HLA-B18, P5-Arg is partially buried in the D-/E-pocket and only exposed its guanidinium group to the solvent, stabilized by a salt bridge with the peptide P3-Asp (Fig. 1b). Polymorphism at position 116 between the two HLA molecules change the architecture of the peptide. The HLA-B18 contains a small Ser116, while the HLA-B37 has a large Phe116. As a result, P9-Phe of the peptide is anchored deeper in the cleft in the HLA-B18 molecule, which

would sterically clash with the P5-Arg conformation observed in HLA-B37. In addition, the salt bridge observed in HLA-B37 between Asp77 and P5-Arg would be lacking in HLA-B18 due to the presence of Ser77. In HLA-B44, P5-Arg was fully exposed to the solvent (Fig. 1c). Owing to the presence of four tyrosine residues (9, 74, 99, 116) in the cleft of HLA-B44 located underneath P5-Arg, Arg cannot be accommodated inside the cleft and adopt a different conformation from the one observed in HLA-B37, that only shares Tyr74. The additional anchor residue observed in the B37-NP$_{338}$ complex could explain the higher stability of this pHLA complex (Supplementary Table 1). These distinct conformations adopted by the NP$_{338}$ peptide, due to HLA polymorphism, changed the surface presented to the TCR. On one hand, the NP$_{338}$ peptide revealed a hydrophobic surface when bound to HLA-B37, while on the other, it was an epitope with central positively charged residues in the cleft of HLA-B18 and HLA-B44.

**Table 2 Conservation of NP$_{338}$ epitope in distinct IAV strains**

| Sequence | Abbreviation | Frequency | | | | | | | | |
|---|---|---|---|---|---|---|---|---|---|---|
| | | IAV—H1N1 | | | IAV—H3N2 | | | Other IAV strains | | |
| | | All | Aust | Vacc | All | Aust | Vacc | pH1N1 All | H5N1 All | H7N9 All |
| *FEDLRVLSF* | *WT* | 0.9 | | | 2.4 | 1.9 | 6.3 | | | |
| *FEDLRV**S**SF* | *L7S* | 96.5 | 93.0 | 100.0 | 1.3 | 1.3 | | 98.5 | 95.9 | 100.0 |
| **FEDLR*L*LSF** | *V6L* | 0.8 | | | 95.6 | 96.9 | 93.8 | | | |
| FED**I**RV**S**SF | L4I+L7S | 0.2 | 4.7 | | | | | | | |
| FEDLRV**SS**L | L7S+F9L | 0.1 | 2.3 | | | | | 0.1 | | |
| FEDLR**IS**SF | V6I+L7S | 0.5 | | | | | | 0.7 | 3.1 | |
| FEDLRV**SG**F | L7S+S8G | 0.8 | | | | | | 0.5 | 1.0 | |
| FEDL**I**V**S**SF | R5I+L7S | | | | | | | | | |
| FEDL**K**VLSF | R5K | 0.1 | | | | | | | | |
| FEDLRV**SS**Y | L7S+F9Y | 0.2 | | | | | | 0.2 | | |
| FEDLRV**T**SF | L7T | 0.1 | | | | | | | | |
| FEDL**GL**LSF | R5G+V6L | | | | 0.1 | | | | | |
| FEDL**KL**LSF | R5K+V6L | | | | 0.3 | | | | | |
| FEDLR**I**LSF | V6I | | | | 0.1 | | | | | |
| FEDLR**L**L**N**F | V6L+S8N | | | | 0.1 | | | | | |
| **S**EDLR**L**LSF | F1S+V6L | | | | 0.1 | | | | | |
| Number of sequences | | 1155 | 43 | 9 | 1264 | 159 | 16 | 858 | 97 | 39 |
| Coverage of selected strains (%) | | 98.1 | 93.0 | 100.0 | 99.4 | 100.0 | 100.0 | 98.5 | 95.9 | 100.0 |

Sequences were obtained from the NCBI Influenza Research Database https://www.ncbi.nlm.nih.gov/genomes/FLU/Database/nph-select.cgi?go=database. Full-length sequences of Australian (denoted Aust), vaccine (denoted Vacc) and pH1N1, H5N1 and H7N9 were obtained and were aligned using the influenza database. Underlined are anchor residues for HLA-B*37:01, and mutations are shown in bold. The first three rows in italics represent sequences chosen for further analysis

We next probed memory NP$_{338}$+CD8+ T cell responses in individuals expressing HLA-B*37:01, HLA-B*44:02, HLA-B*44:03, or HLA-B*18:01 (Fig. 1e–g, Supplementary Table 3) by stimulating their peripheral blood mononuclear cells (PBMCs) with a pool of NP$_{338}$ peptides corresponding to the main variants in the circulating influenza A strains (Table 2). Following 10–16 days of peptide stimulation, we detected robust CD8+ T cell responses by interferon-γ (IFNγ) and tumor necrosis factor-α (TNFα) production as well as CD107a expression in the majority (4/5) of HLA-B37+ donors (Fig. 1f, g), with a high level of polyfunctionality (Fig. 1g). However, NP$_{338}$+ CD8+ T cell responses were undetectable in all (5/5) HLA-B*44:02, HLA-B*44:03, or HLA-B*18:01 individuals (Fig. 1f, g), despite their CD8+ T cells having responded strongly to non-specific stimulation by phorbol 12-myristate 13-acetate or Concanavalin A (Fig. 1f).

Overall, these data suggest that HLA-B37, but not HLA-B44 or HLA-B18, represents the HLA restriction for the NP$_{338}$ peptide. The lack of NP$_{338}$+CD8+ T cell responses in the context of HLA-B44 and HLA-B18 might be due to the distinct conformations, or lower stability, of NP$_{338}$ as compared to HLA-B37.

**Cross-reactive NP$_{338}$+CD8+ T cells towards NP$_{338}$ variants.** Having shown robust NP$_{338}$+CD8+ T cell responses in HLA-B37 donors, we next determined the level of cross-recognition towards the main NP$_{338}$ variants occurring in IAVs. Our conservation analysis of the NP$_{338}$ viral peptides across vaccine and circulating IAV strains, including pH1N1, H5N1, and H7N9, found 16 natural variants of NP$_{338}$ peptide (Table 2), with the most common mutations occurring at position (P) 5, 6, and 7 of the peptide. Two major variants; FEDLRVSSF (NP$_{338}$-L7S) and FEDLRLLSF (NP$_{338}$-V6L) were dominant in H1N1, pH1N1, H5N1, and H7N9 or H3N2 strains, respectively, representing 93–100% of the distinct strains and were thus selected in addition to the previously published (WT) peptide for further analysis.

To assess the cross-reactive potential of HLA-B37-NP$_{338}$+CD8+ T cells, pooled or variant-specific NP$_{338}$ T cell lines were generated using PBMCs from HLA-B37+ donors. We also generated HLA-B37-NP$_{338}$ and mutant HLA-B37-NP$_{338}$-V6L and HLA-B37-NP$_{338}$-L7S tetramers. The ability to recognize, and functionally respond, to each of the NP$_{338}$ variant was then assessed using tetramers and an intracellular cytokine staining assay, respectively (Fig. 2). Staining with all three pHLA tetramers and functional assays showed similar proportions of CD8+ T cell staining/functional levels in all WT or variant-specific NP$_{338}$-T cell lines (Fig. 2a, c), suggesting that the majority of NP$_{338}$-specific CD8+ T cell lines were able to cross-recognize (Fig. 2a, c) and respond to (Fig. 2b, c) each of the three distinct peptides, indicating broad cross-reactivity towards these epitopes (Fig. 2b, c). Interestingly, the NP$_{338}$-V6L tetramer showed a slightly stronger staining than the WT and NP$_{338}$-L7S tetramers (Fig. 2a).

CD8+ T cell lines typically recognized and responded most prominently to the peptide they were generated against, with the lowest level of cross-reactivity being observed by the NP$_{338}$-V6L-specific CD8+ T cell lines, when stimulated with the NP$_{338}$-WT (21.45 ± 18.86% by tetramer, 20.98 ± 19.80% by IFNγ, and 18.88±18.90% by TNFα) and NP$_{338}$-L7S variant (16.31 ± 14.47% by tetramer, 13.63 ± 12.04% by IFNγ, and 13.04 ± 11.65% by TNFα). These data suggest that a high level of cross-reactivity across different variants exists, although some variant-specific reactivity can be also detected. These data highlight that such cross-reactive CD8+ T cells may offer protection against H1N1, H5N1, H7N9, and possibly other novel IAV strains.

To further assess whether the WT NP$_{338}$ peptide elicited CD8+ T cell responses of high functional avidity, CD8+ T cell lines were generated against the WT NP$_{338}$ peptide (n = 3), and their functional response was assessed towards decreasing concentrations of the WT NP$_{338}$ peptide (Fig. 2d). All data were normalized to the maximal response (when stimulated with maximum peptide). High and stable functional avidity towards NP$_{338}$ (IFNγ, TNFα, and CD107a) was observed in cell lines derived from all three donors (Fig. 2d).

Our data show that HLA-B37+ individuals display polyfunctional and highly cross-reactive CD8+ T cell response to the NP$_{338}$ epitope and its major variants.

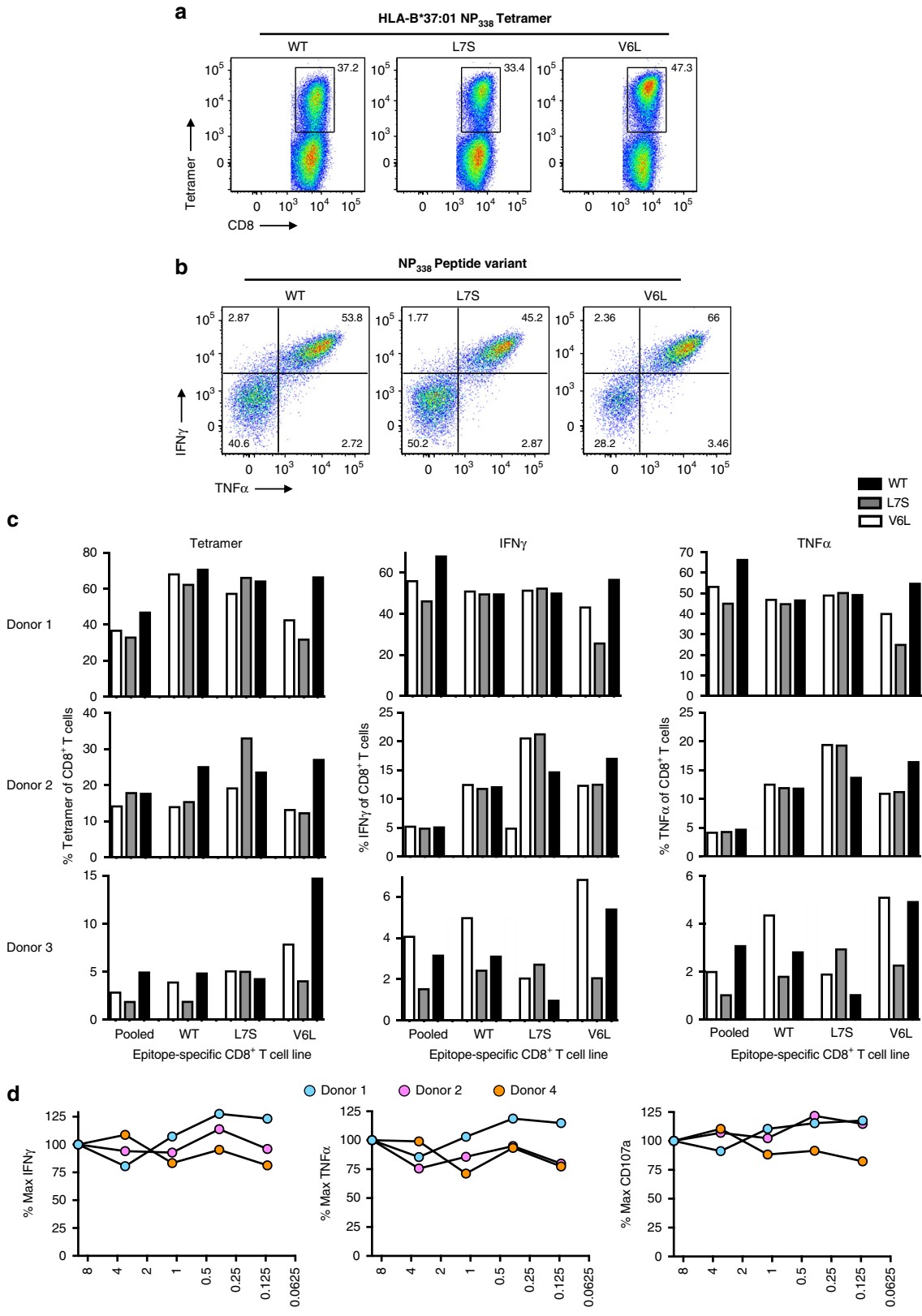

**NP$_{338}$ variants are molecular mimics of the WT NP$_{338}$ epitope**. To understand the molecular basis of HLA-B37-NP$_{338}$+CD8+ T cell cross-reactivity, we solved the structures of two major NP$_{338}$ variants, namely, NP$_{338}$-V6L and NP$_{338}$-L7S in complex with HLA-B37 (Fig. 3; Supplementary Table 2). Both peptides bound HLA-B37 in an extended conformation with the canonical P2-Glu and P9-Phe[49], while P5-Arg acted as a secondary anchor residue (Fig. 3a, b). Superimposition of the HLA-B37-NP$_{338}$ binding cleft with either HLA-B37-NP$_{338}$-V6L (Fig. 2c) or HLA-B37-NP$_{338}$-L7S (Fig. 2d) revealed a similar conformation of the HLA-binding cleft (r.m.s.d. of 0.08 Å). In addition, the peptides adopted a similar conformation as the NP$_{338}$ peptide with an r.m.

**Fig. 2** $NP_{338}$-specific $CD8^+$ T cell lines cross-react with variant $NP_{338}$ peptides. PBMCs from HLA-B*37:01$^+$ donors ($n = 4$) were stimulated with a pool (10 µM each, 30 µM total) or individual (10 µM) $NP_{338}$ peptides for 10–11 days and cross-reactivity was assessed by tetramer staining and ICS assays. Gating strategy: lymphocytes, singlets, live$^+$, $CD3^{mid\text{-}high}$, $CD8^+$ T cells; as per Supplementary Figure 2a, b. **a** Representative dot plots of a pooled $NP_{338}$-specific $CD8^+$ T cell line stained with each $NP_{338}$ variant tetramers. **b** Representative dot plots of $IFN\gamma^+TNF\alpha^+$ production by pooled $NP_{338}$-specific $CD8^+$ T cells. **c** Summary of tetramer staining, $IFN\gamma$ (minus no peptide control), and $TNF\alpha$ (minus no peptide control) production by pooled or variant-specific $NP_{338}$-specific $CD8^+$ T cell lines in Donors 1–3. **d** Summary of $IFN\gamma$, $TNF\alpha$, and CD107a production, minus no peptide control, of WT-specific $CD8^+$ T cell lines from Donors 1, 2, and 4 in response to concentrations ranging between 10 µM and 0.125 µM of the WT-$NP_{338}$ peptide. Values are represented as a proportion of maximum tetramer staining or cytokine production against the line-specific peptide

s.d. of 0.12 and 0.09 Å, respectively (Fig. 3e, f). The hydrophobic P6-Val residue sits near the $\alpha 2$-helix and interacts with Val152, Gln155, and Asp156 of the HLA (Fig. 3g). The substitution of P6-Val for P6-Leu in the H3N2 was readily accommodated in the HLA-B37 cleft and did not change either the peptide or HLA conformation (Fig. 3h). The P7-Leu residue sits above Trp147, with its side chain interacting with Ala150, Val152, and Lys146 (Fig. 3i). The substitution from P7-Leu to P7-Ser did not impact the conformation of either the peptide or HLA molecule (Fig. 3j). We also showed that the two $NP_{338}$ variants could stabilize HLA-B37 as efficiently as the $NP_{338}$ WT peptide (Supplementary Table 1).

Overall, the two major variants of the $NP_{338}$ epitope adopted the same conformation as the $NP_{338}$ epitope and represent molecular mimics of the WT peptide, which might favor T cell cross-reactivity.

**Key TCRs cross-recognize $NP_{338}$ variants directly ex vivo.** To investigate HLA-B37-$NP_{338}^+CD8^+$ T cell cross-reactivity at a molecular level, PBMCs from healthy HLA-B37$^+$ donors ($n = 3$) were stained with WT and variant HLA-B37-$NP_{338}$ tetramers individually, magnetically enriched, and single-cell sorted. TCRαβ repertoire was determined using a multiplex RT-PCR[37]. As expected, prominent $CD8^+$ T cell memory pools could be detected directly ex vivo using $NP_{338}$, $NP_{338}$-V6L, and $NP_{338}$-L7S tetramers (Fig. 4a), with magnetic enrichment increasing the proportion of $NP_{338}$-specific $CD8^+$ T cells >100-fold. $CD8^+$ T cells directed at the HLA-B37-$NP_{338}$ epitopes in healthy donors were of the memory phenotypes characterized by $CD27^{hi}$ and $CD45RA^{low\text{-}med}$ expression (Fig. 4a). Interestingly, $NP_{338}$-V6L-specific $CD8^+$ T cells were found at the highest average frequency, at 1 in every $5.97 \times 10^{-4}$ $CD8^+$ T cells, followed by $NP_{338}$ and then $NP_{338}$-L7S-specific $CD8^+$ T cells found at frequencies of 1 in $2.31 \times 10^{-4}$ and $1.86 \times 10^{-4}$ $CD8^+$ T cells, respectively (Fig. 4b).

The TCRαβ repertoire across all three donors (Fig. 5, Supplementary Table 4) was entirely private, meaning no shared TCRαβs were detected between the donors, and of comparable diversity, comprising $13.67 \pm 5.77$, $11.67 \pm 0.58$, and $10.33 \pm 2.08$ distinct TCRαβ clonotypes (Donors 1–3, respectively). A TRBV19 bias was detected in all donors ($19.01 \pm 18.54\%$, $39.88 \pm 27.32\%$, and $37.36 \pm 33.49\%$ in Donors 1–3, respectively), while Donor 1 displayed an additional TRBV27 bias ($26.01 \pm 16.12\%$; Fig. 5, Supplementary Table 4). Interestingly, only one or two high-frequency but unique clonotypes were able to cross-recognize all three $NP_{338}$ variants, and $69.24 \pm 16.65\%$, $44.25 \pm 34.87\%$, and $53.45 \pm 5.76\%$ cross-reactivity was observed in Donors 1–3, respectively. Interestingly, each of these donors utilized different pairings to achieve universal cross-reactivity. Donor 1 favored TRAV14/TRBV20-1 and TRAV29/TRBV27, Donor 2 TRAV35/TRBV19, and Donor 3 TRAV23/TRBV27 and TRAV21/TRBV9 pairings (Supplementary Table 4). In addition, TRBV19$^+$ cross-reactive TCR clonotypes were detected in all donors.

With respect to the CDR3α/β length, 9-mer CDR3α and 9-10-mer CDR3β loops were preferentially used for the recognition of $NP_{338}$ (Fig. 5b, c). Together, these data highlight the potential of

common HLA-B37-$NP_{338}^+CD8^+$ TCRαβ clonotypes to provide protection against distinct IAV strains.

The frequency of cross-reactivity was increased ($81.35 \pm 19.72\%$) following in vitro expansion with the WT $NP_{338}$ peptide (Supplementary Fig. 1). Donor 3's TCRαβ repertoire was more restricted following in vitro amplification comprising fewer distinct clonotypes ($7.33 \pm 2.08$) and displayed a stronger TRBV19 bias (Supplementary Fig. 1ab). Again, 7–10-mer CDR3α and 8–9-mer CDR3β lengths were preferred in the recognition of $NP_{338}$ (Supplementary Fig. 1c). A single TCRαβ clonotype (termed EM2: TRAV30/TRBV19, CDR3α: CGTERSGGYQKVTF, and CDR3β: CASSMSAMGTEAFF) was able to cross-react with all three $NP_{338}$ peptides, therefore providing universal HLA-B37-$NP_{338}^+CD8^+$ recognition.

Overall, despite entirely private TCRαβ repertoires being detected within each individual, a high level of inter-variant cross-reactivity was achieved in each donor, facilitated by key TCRs.

**Biased TRBV19$^+$ TCR chains mainly contacts the HLA-B37.** To further understand the molecular mechanism underlying T cell cross-reactivity, we then investigated the cross-reactive clonotype described above (EM2 TCR; TRAV30/TRBV19, Supplementary Table 4) that was able to recognize $NP_{338}$ and its two most common variants. We solved the structure of the EM2 TCR in complex with HLA-B37-$NP_{338}$ (Supplementary Table 2, Fig. 6). The EM2 TCR docked centrally onto HLA-B37-$NP_{338}$ with a 66° angle (Fig. 6b), falling in the range previously observed for TCR-pHLA-I (average of 63°)[50]. The buried surface area (BSA) of the complex was of ~1800 Å$^2$, also falling in the previously observed range for TCR-pHLA-I (average of 1900 Å$^2$)[50]. The EM2 TCR α-chain contributes to 40% of the contact surface (Fig. 6c), while the biased TRBV19$^+$ β-chain contributes to 60% of the BSA, providing a direct basis for the TRBV19 bias observed in HLA-B37$^+$ donors in response to $NP_{338}$ and its variants. The peptide contributed to 25% of the pHLA BSA, with all six solvent-exposed residues contacted by the EM2 TCR. CDR2β (27%), CDR3α (23%), and CDR3β (20%) are the highest contributors to the BSA, followed by CDR1α, CDR2α, and CDR1β (6–8% each), as well as the framework (FW) residues of the β-chain (9%) (Fig. 6c).

All of the CDR loops of the EM2 TCR contacted the HLA-B37 molecule (Fig. 6d), with the addition of two FW residues from the β-chain (Gln67β and Lys83β). From the EM2 α-chain, the CDR1/2 loops contributed to ~10% of the TCR-HLA BSA and 16% for the CDR3α (Supplementary Table 5). The CDR1/2α loops made a contact with the region before the hinge of the HLA α2-helix (residue 151–158, Fig. 6f), while the CDR3α loop contacted the N-terminal part of the HLA α1-helix (residues 58 and 62, Fig. 6g). Within the β-chain, the CDR1 made only few contacts with the HLA (~2% TCR-HLA BSA), and the contacts were driven by CDR2β (34%), CDR3β (14%), as well as FWβ (11%) (Fig. 6d). The CDR3β loop contact with the HLA focused mostly on Gln155, which has its side chain sandwiched by the Tyr30α (CDR1α) and the $^{110}$SAMGT$^{114}$ motif from the CDR3β loop (Supplementary Table 5, Fig. 6h). The FW residue Lys83β formed

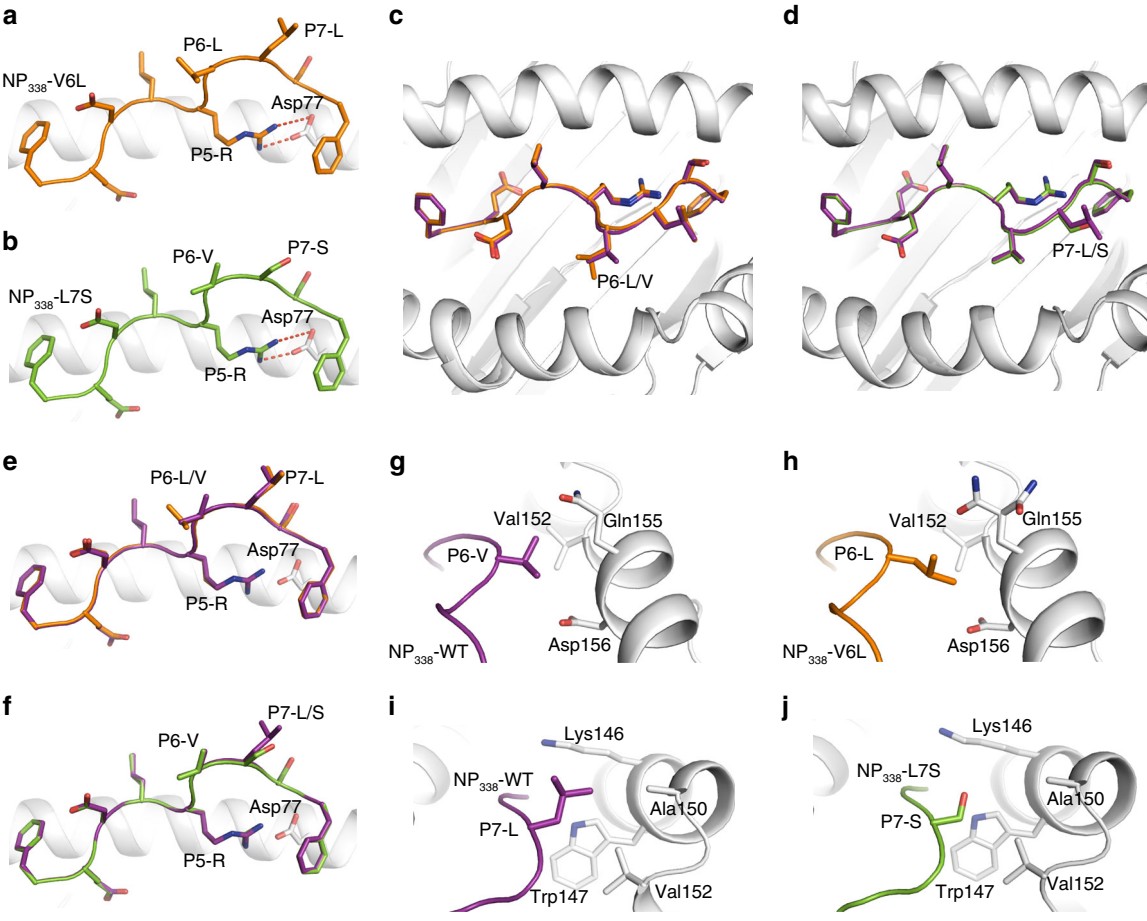

**Fig. 3** Structures of HLA-B37 bound to NP$_{338}$ and its two variants. Structure of the HLA-B37 (white cartoon) presenting either **a** NP$_{338}$-V6L (orange) or **b** NP$_{338}$-L7S (green). **c**, **d** Top view or **e**, **f** side view of the structural overlay of HLA-B37 (white cartoon) presenting the NP$_{338}$ (purple) with either **c**, **e** NP$_{338}$-V6L (orange) or **d**, **e** NP$_{338}$-L7S (green). **g–j** Zoomed in view on the HLA-B37 interaction with residue at position 6 (**g**, **h**) and position 7 (**i**, **j**) of the NP$_{338}$ (purple), NP$_{338}$-V6L (orange) and NP$_{338}$-L7S (green) peptides. The red dashed lines represent salt bridges between residues

a network of salt bridge with the HLA-B37 Glu76 and CDR1β Asp30, hovering above P8-Ser (Fig. 6i). The CDR2β loop form a hairpin loop that sat above the HLA α1-helix covering a large stretch of the HLA helix spanning from residue 65 to 79 (Fig. 6b, d, j and Supplementary Table 5). The residue Gln67β from the FW, located downstream of the CDR2β loop was also contacting the α1-helix of the HLA-B37 molecule. Gln57β, Asn60β, Asp61β, and Gln67β were forming a large hydrogen network with the HLA-B37 molecule.

Overall, CDR2β and FWβ make extensive interaction with the HLA-B37 molecule, providing a basis for the strong bias TRBV19+ usage observed in the donors.

**EM2 TCR contacts the NP$_{338}$ peptide via a hydrophobic network.** The peptide is predominantly contacted by CDR3α (47%) and CDR3β (26%), as well as by CDR1β (17%) and CDR2β (10%) (Fig. 6b, e, Supplementary Table 5). The motif SGG within the CDR3α loop, with small or no side chain, allow the loop to be in close proximity and lodge itself between the side chains of P1-Phe and P3-Asp (Fig. 7a). The proximity of the loop allows the Gln113α to form hydrogen bonds with P3-Asp of the peptide and hydrophobic interaction with P4-Leu. Therefore, CDR3α loop forms a lid-like structure that covers the N-terminal part of the peptide (Supplementary Table 5). Within the EM2 TCR β-chain, all the CDR loops were contacting the peptide C-terminal section from residue P4-Leu to P8-Ser. P4-Leu was contacted by Gln57β from the CDR2β loop, while CDR1β loop was interacting with

both P7-Leu and P8-Ser that form a hydrogen bond with Asp30β (Supplementary Table 5). P6-Val and P7-Leu were both contacted by the CDR3β loop that sat on top of the central region of the peptide and form Van der Waals interaction. The [109]MSAM[112] motif from the CDR3β loop could hover on the central part of the peptide due to the small side chains of the [110]SA[111] residues. The CDR3β loop was thus making contacts with P6–P7 residues of the peptide mainly via Van der Waals bonds (Supplementary Table 5).

Interestingly, the comparison of HLA-B37-NP$_{338}$ in its free and liganded state with the EM2 TCR revealed that the peptide (r.m.s. d. of 0.2 Å) and HLA-B37 (r.m.s.d. of 0.3 Å) did not change conformation upon TCR binding (Fig. 7c). The hydrophobic nature of the interaction between the EM2 TCR and the NP$_{338}$ peptide is complementary with the conformation of the NP$_{338}$ peptide bound to HLA-B37 that presents only the hydrophobic residue to the TCR. This might explain the lack of response for NP$_{338}$ when presented by both HLA-B18 and HLA-B44 molecules, in which the peptide presents its charged P5-Arg to the TCR (Fig. 1b, c).

Altogether, the structure of the EM2 TCR–HLA-B37-NP$_{338}$ complex revealed that the TCR is using hydrophobic interaction from its CDR3 loops to engage with the all stretch of the NP$_{338}$ peptide.

**EM2 TCR affinity for NP$_{388}$-V6L is higher than for NP$_{388}$-WT.** We next evaluated the ability of the EM2 TCR to recognize the NP$_{338}$ and its two major variants (NP$_{338}$-V6L and NP$_{338}$-L7S)

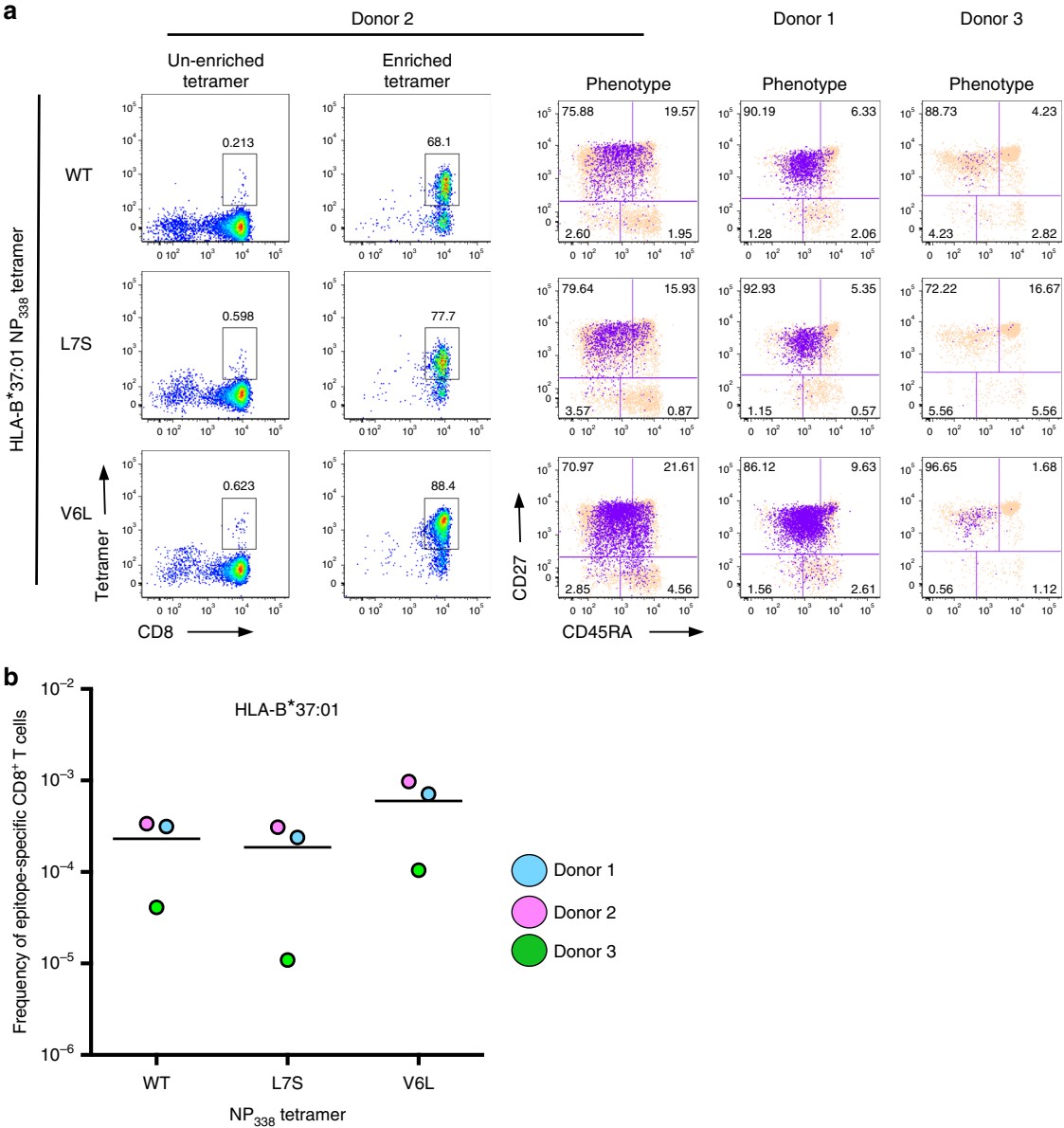

**Fig. 4** HLA-B37[+] NP$_{338}$-specific CD8[+] T cells recognize NP$_{338}$ variants directly ex vivo. PBMCs from HLA-B*37:01[+] donors ($n = 3$) were tetramer stained individually with each variant NP$_{338}$ tetramer conjugated to PE and assessed via flow cytometry. Samples were gated on lymphocytes, singlets, CD3[+]live[+]dump[−] cells; as per Supplementary Figure 2c. **a** Representative dot plots of tetramer staining of CD3[+] cells prior to and following enrichment (numbers respond to a proportion of tetramer[+] of CD8[+] T cells) and CD27/CD45RA phenotype of tetramer[+] CD8[+] T cells (purple) superimposed on unenriched CD8[+] T cells. **b** Frequency of NP$_{338}$ variant-specific CD8[+] T cells in HLA-B*37:01[+] donors

using surface plasmon resonance (Fig. 7d–f; Supplementary Table 6). The EM2 TCR bound to the HLA-B37-NP$_{338}$ complex with a $K_d$ of ~130 μM (Fig. 7d), displaying a low affinity for an antiviral CD8[+] T cell, the average being 35 μM[50]. The EM2 TCR bound to NP$_{338}$-L7S with a similar affinity to the one observed for the NP$_{338}$ bound to HLA-B37 ($K_d$ ~ 150 μM, Fig. 7e). Interestingly, the EM2 TCR affinity was increased by four-fold for the NP$_{338}$-V6L ($K_d$ ~ 30 μM, Fig. 7f) compared to NP$_{338}$ when bound to HLA-B37.

The structures of each NP$_{338}$ and variants in complex with the HLA-B37 molecule (Fig. 3, Supplementary Table 2) allowed us to understand the impact of the NP$_{338}$ mutation onto the EM2 TCR recognition. Overlay of the HLA-B37-NP$_{338}$-L7S with the EM2 TCR-HLA-B37-NP$_{338}$ structures showed that the smaller P7-Ser would be readily accommodated by the EM2 TCR without structural changes (Fig. 7g). This might explain the similar

affinity observed for the EM2 TCR towards both peptides ($K_d$ ~ 130–150 μM). In a similar fashion, an overlay of the HLA-B37-NP$_{338}$-V6L with the EM2 TCR-HLA-B37-NP$_{338}$ structures (Fig. 7h) shows that the larger side chain of the P7-Leu might create additional contact with the CDR3β loop and might in turn provide a basis for the improved affinity observed for the EM2 TCR towards the HLA-B37-NP$_{338}$-V6L ($K_d$ ~ 30 μM).

Thus our data demonstrate that CD8[+] T cell cross-reactivity in the context of NP$_{338}$ peptide and its variants is favored by similar conformations adopted by the epitopes.

**Key TCRs are cross-reactive towards A1-NP$_{44}$ variants.** To investigate whether cross-reactivity and thus protection against distinct IAV strains is NP$_{338}$-specific or occurs towards other IAV-specific epitopes, we assessed CD8[+] T cell responses towards

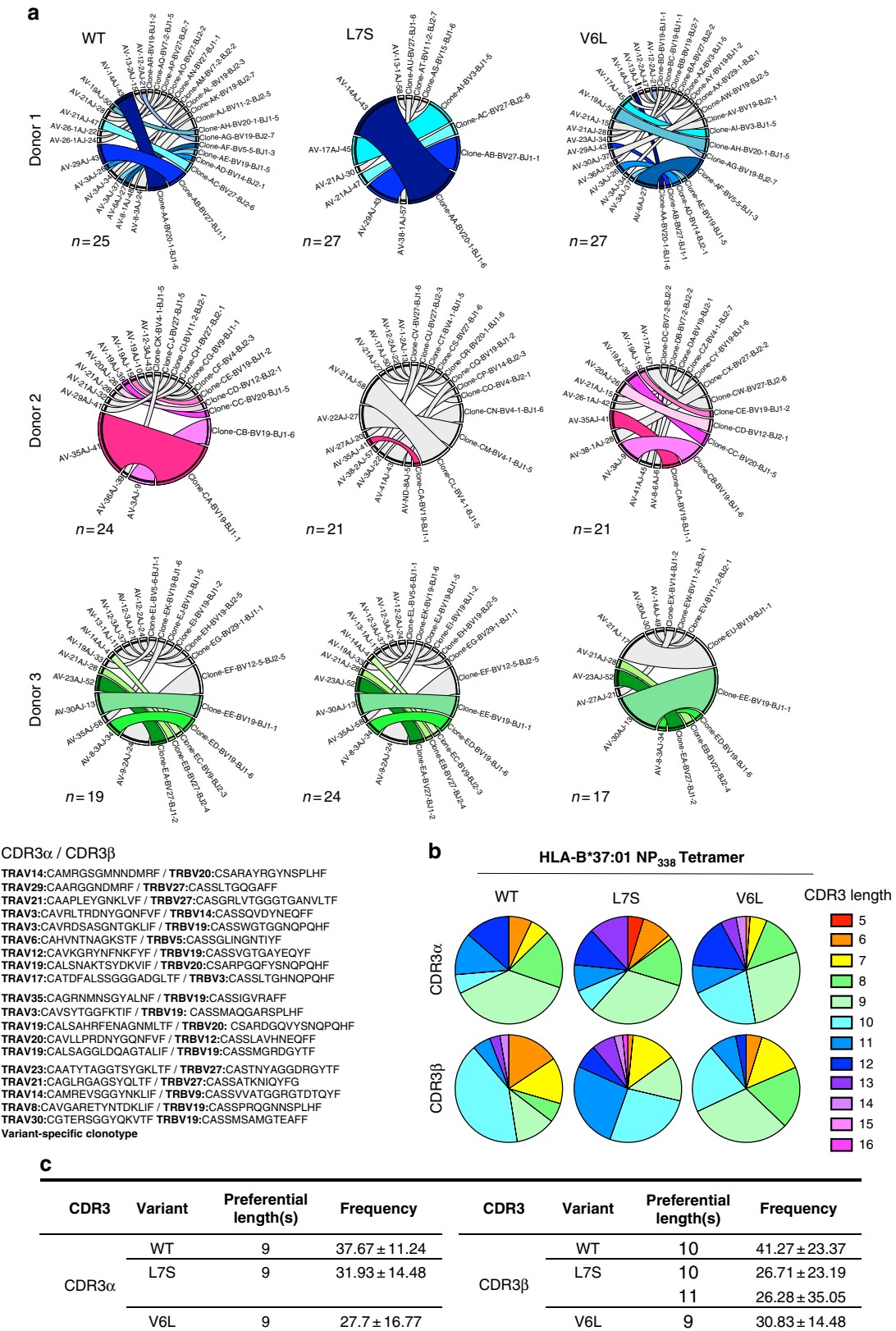

**Fig. 5** Key TCRs cross-recognize NP$_{338}$ variant peptides directly ex vivo. PBMCs from HLA-B*37:01$^+$ donors (Donors 1–3) were tetramer stained individually with each of the NP$_{338}$ tetramers conjugated to PE, enriched, and single-cell sorted on lymphocytes, singlets, CD3$^+$live$^+$dump$^-$ cells, CD8$^+$tet $^+$ cells; as per Supplementary Figure 2c. The TCRαβ repertoire was determined using a multiplex RT-PCR. **a** Graphical representation of the TCRαβ repertoire used by each of three donors for the recognition of variant NP$_{338}$ peptides. **b** Average CDR3α and CDR3β length used for the recognition of NP$_{338}$ from HLA-B*37:01$^+$ donors. **c** Preferred CDR3α and CDR3β length of NP$_{338}$-specific cells isolated from HLA-B*37:01$^+$ donors in the recognition of variant NP$_{338}$ peptides

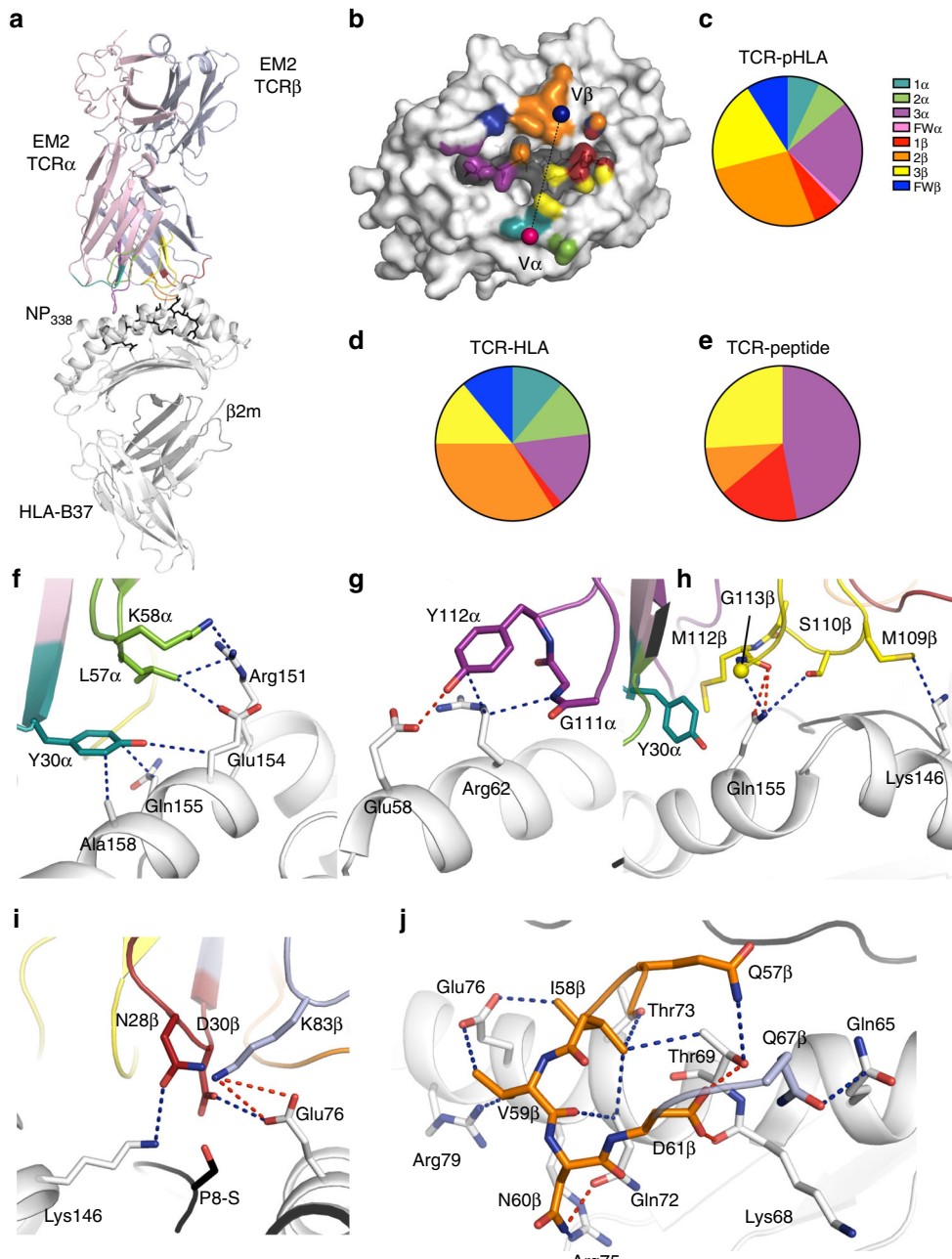

**Fig. 6** Structure of EM2 TCR in complex with the HLA-B37-NP$_{338}$. **a** Overview of the EM2 TCR (α-chain in pink and β-chain in blue) bound to the HLA-B37-NP$_{338}$ complex (HLA in white and peptide in black). **b** The EM2 TCR-binding footprint on the HLA-B37 surface (white) and NP$_{338}$ peptide (gray surface) is represented by the atomic contact made by each TCR residues colored in teal, green, and purple for the CDR1, 2, and 3 of the α-chain; red, orange, and yellow for the CDR1, 2, and 3 of the β-chain; and pink and blue when contacted by the framework residue of the α- and β-chain, respectively. The pie charts represent the buried surface area at the interface between **c** the TCR with the HLA-B37-NP$_{338}$ or with **d** HLA-B37 alone or **e** with the peptide alone. The different colors represent the TCR gene segment as per **b**. Interaction of the EM2 TCR with the HLA-B37 molecule (white) are represented for **f** CDR1α and CDR2α; **g** CDR3α; **h** CDR3β; **i** CDR1β and FWβ; and **j** for CDR2β. The same color scheme is used as per **b** for the different segments of the TCR. The sphere represents the Cα atom of glycine residue, the blue dashed lines represent hydrophobic interaction, while the red dashed lines indicated hydrogen bond or salt bridges

the HLA-A1-restricted NP$_{44}$ peptide (Fig. 8). Conservation analysis revealed that there were two major variants of the peptide, the CTELKLSDY (NP$_{44}$) and the S7N mutant CTELKLNDY (NP$_{44}$-S7N), covering 34–100% of H3N2, H7N9, H5N1, and H1N1 virus strains (Supplementary Table 7). To investigate cross-reactivity towards these variant NP$_{44}$ peptides, PBMCs from healthy HLA-A1$^+$ donors ($n = 3$) were tetramer stained with each of the NP$_{44}$ variants directly ex vivo and magnetically enriched and single-cell sorted (Fig. 8a). The TCRαβ repertoire

was then determined using multiplex RT-PCR (Fig. 8b). Two of the three donors (Donors 1 and 20) had both NP$_{44}$- and NP$_{44}$-S7N-specific CD8$^+$ T cell populations. Interestingly, Donor 21 had only a NP$_{44}$-specific CD8$^+$ T cell population, with no detectable S7N-specific CD8$^+$ T cell population (Fig. 8b). A predominant TRAV8-2$^+$ TCRα chain expressing a CDR3α sequence CVVSDRNFNKFYF paired with multiple distinct TCRβs and was detected in two out of three donors (Supplementary Table 8). Similar to the NP$_{338}$ system, specific clonotypes

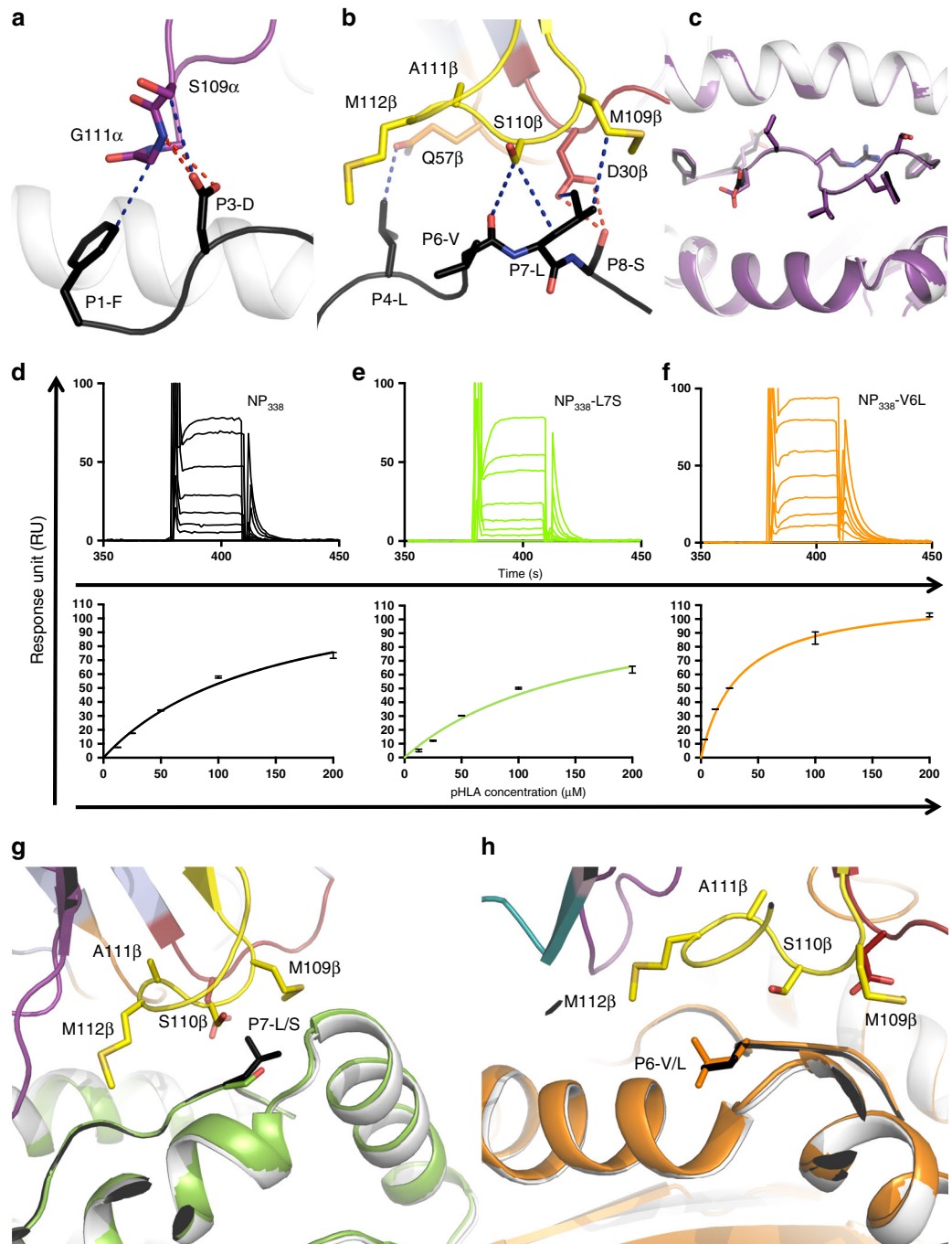

**Fig. 7** EM2 TCR recognizes the structurally similar NP$_{338}$ variants. **a** Interaction between the EM2 TCR CDR3α loop (purple) with the NP$_{338}$ peptide (black). **b** Interaction of the NP$_{338}$ peptide (black) with the CDR1β (red), CDR2β (orange), and the CDR3β loops (yellow). The blue and red dashed lines indicate Van der Waals or hydrogen bonds, respectively. **c** Top view of the structural overlay of the HLA-B37-NP$_{338}$ free (purple) or bound to the EM2 TCR (white for HLA and black for the peptide). **d–f** Surface plasmon resonance sensorgrams (top panel) and equilibrium binding curves (bottom panel) of the EM2 TCR for the HLA-B37 presenting the NP$_{338}$ (**d**, in black), NP$_{338}$-L7S (**e**, in green), or NP$_{338}$-V6L (**f**, in orange). The experiment was done twice in duplicate. **g** Overlay of the EM2 TCR-HLA-B37-NP$_{338}$ (HLA in white and peptide in black) with the HLA-B37-NP$_{338}$-L7S (green) structures. The EM2 TCR is colored as per Fig. 6, with the CDR3β loop in yellow. **h** Overlay of the EM2 TCR-HLA-B37-NP$_{338}$ (HLA in white and peptide in black) with the HLA-B37-NP$_{338}$-V6L (orange) structures

isolated from Donor 1 could recognize both NP$_{44}$ variants, while the remainder of the repertoire was variant specific (Fig. 8b). Interestingly, the TCRαβ repertoire of Donor 20 was entirely NP$_{44}$ variant specific. We previously solved the structures of both NP$_{44}$ and NP$_{44}$-S7N peptides in complex with the HLA-A1 molecule[23]. In similar fashion as HLA-B37 and the NP$_{338}$ variants, the NP$_{44}$ variants adopted similar conformation in the cleft

of HLA-A1 molecule. This suggests that here too molecular similarity might underpin CD8$^+$ T cell cross-reactivity and favor the recognition of multiple IAV strains.

Overall, these data show that TCRαβ cross-reactivity is not limited to the recognition of HLA-B37-NP$_{338}$ and highlight the importance of cross-reactive CD8$^+$ T cells in the recognition of virus-infected cells and thus their potential to protect against distinct IAV strains.

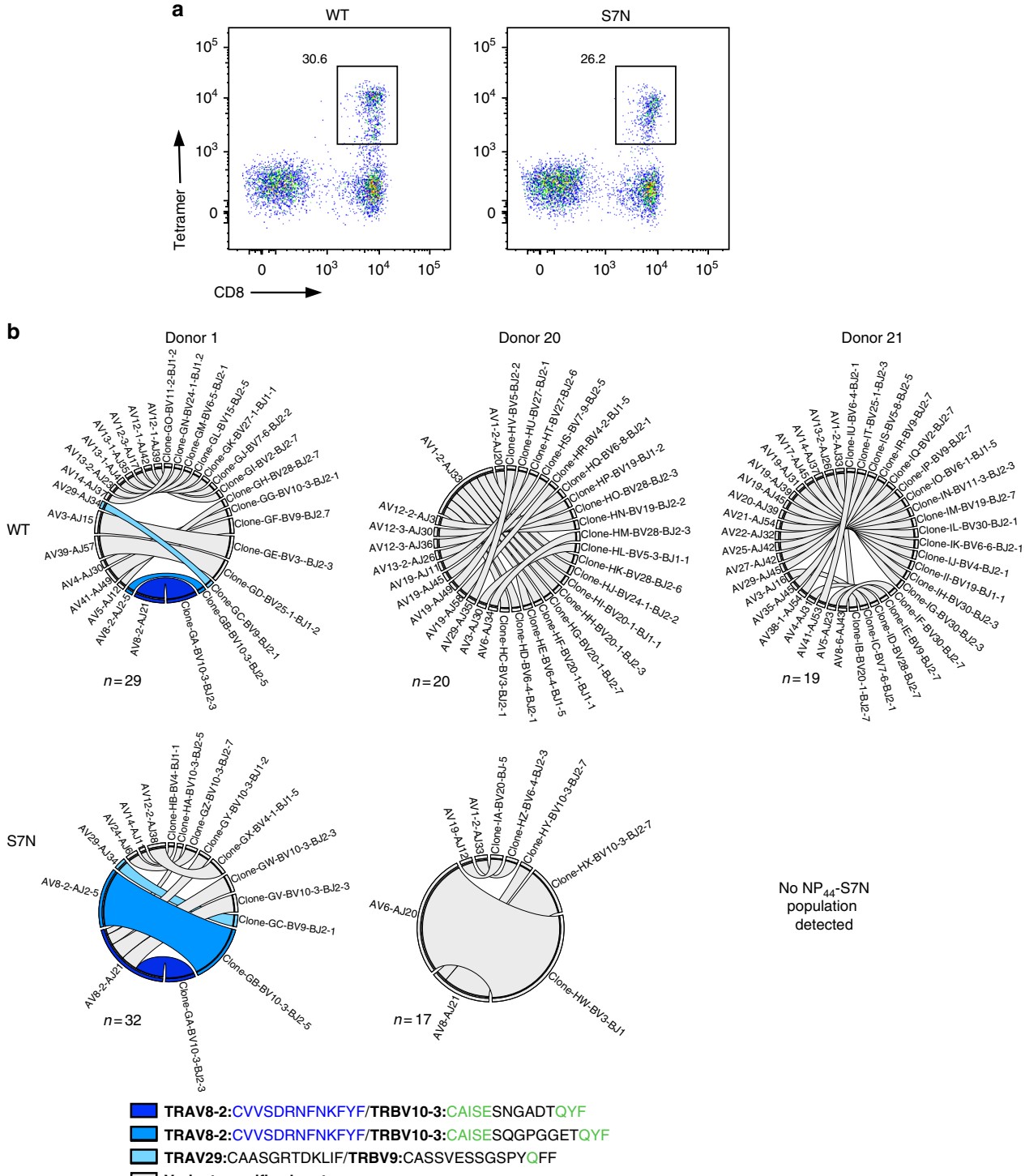

**Fig. 8** TCRαβ clonotypes can recognize HLA-A*01:01-restricted NP$_{44}$ variants. PBMCs from HLA-A*01:01$^{+}$ donors (Donors 1, 20 and 21) were tetramer stained individually with each of the NP$_{44}$ tetramers conjugated to PE. Samples were enriched, surface stained, and single-cell sorted on lymphocytes, singlets, CD3$^{+}$live$^{+}$dump$^{-}$ cells, CD8$^{+}$tet$^{+}$ cells, as per Supplementary Figure 2c, and the TCRαβ repertoire was determined using a multiplex RT-PCR. **a** Representative dot plots of NP$_{44}$ tetramer staining of CD3$^{+}$ T cells following magnetic enrichment. Numbers are represented as a proportion of tetramer$^{+}$ of CD8$^{+}$ T cells. **b** Summary of the TCRαβ repertoire used by each of the three donors for the recognition of the variant NP$_{44}$ peptides

## Discussion

There is great interest in the development of a universal CD8$^{+}$ T cell-mediated vaccine towards IAV. However, to realize this, first a thorough understanding of prominent human influenza-specific CD8$^{+}$ T cell epitopes is needed. We have previously shown that NP is the most immunogenic antigen in both HLA-A*02:01-positive[51]

and -negative individuals[39]. However, as the conservation of the known influenza epitopes varies between 16% and 56%, depending on ethnicity[23], understanding inter-epitope cross-recognition across diverse IAV strains is needed. In this study, we have used the variable NP$_{338}$ and NP$_{44}$ peptides to dissect CD8$^{+}$ T cell recognition of different peptide variants and thus distinct IAV strains.

$NP_{338}$ has two major variant peptides, covering ~93% ($NP_{338}$-L7S) and ~96% ($NP_{338}$-V6L) of H1N1 and H3N2 viruses, respectively. Although $CD8^+$ T cell responses are highly peptide specific[34], they can tolerate variations within $CD8^+$ T cell peptides[22,29,30,34]. We identified functional cross-reactivity towards the variant $NP_{338}$ peptides, albeit to different levels. Using a multiplex RT-PCR for the detection of the TCRαβ repertoire directly ex vivo, we confirmed that $CD8^+$ T cells were cross-reactive towards these distinct $NP_{338}$ variants, at a clonotypic level, with ~55% TCRαβ repertoire being capable of recognizing multiple $NP_{338}$ variants. Interestingly, unique clonotypes with distinct TRAV–TRBV pairings in each individual recognized all three $NP_{338}$ variants. Furthermore, this cross-reactivity was increased following in vitro amplification, potentially suggesting that activation increases expansion of cross-reactive clones, allowing certain TCRαβ clonotypes to recognize distinct viral variants and potentially prevent viral escape. Additionally, we showed that the importance of cross-reactive TCRαβ repertoires is not limited to the recognition of $NP_{338}$ and that cross-reactive $CD8^+$ TCRαβ can recognize HLA-A1-restricted $NP_{44}$ variant peptides.

To understand the molecular basis of cross-reactive TCRαβ recognition, we solved the structure of $NP_{338}$ and its variants bound to the HLA-B37 molecule, as well as the structure of a cross-reactive TCR in complex with HLA-B37-$NP_{338}$. First, our data show that the two $NP_{338}$ variants were structural mimics of the WT peptide. Second, the $NP_{338}$ peptide conformation was not altered by TCR engagement, and therefore the conserved structure of the $NP_{338}$ variants favored TCR cross-reactivity, a mechanism that is thought to underpin cross-reactivity and alloreactivity more broadly[45,52,53]. The structures also demonstrated that only HLA-B37, and not HLA-B18 nor HLA-B44, binds the $NP_{338}$ with a secondary anchor residue (P5-Arg), revealing the hydrophobic surface of the $NP_{338}$ peptide, which might help cross-reactivity[36,54]. This hydrophobic surface of the $NP_{338}$ was disrupted when the peptide was presented by HLA-B44 or HLA-B18 molecules, which exposed the charged P5-Arg, and for which no T cell recognition was observed.

Together, these data imply that $HLA-B37^+$ donors may have a level of protection against H1N1, pH1N1, H3N2, H5N1, H7N9, and possibly novel IAV strains. Furthermore, the findings highlight the potential of inducing inter-strain cross-protection by the inclusion of a single peptide variant in a $CD8^+$ T cell-mediated vaccine, suggesting that vaccine epitope identification will no longer need to focus solely on conserved epitopes for universal inter-strain cross-protection.

Diverse and cross-reactive TCR repertoires are important for viral control and preventing viral escape[27,29], and we thus dissected the importance of TCRαβ diversity in the recognition of the variant $NP_{338}$ peptides. Our data show that cross-recognition of distinct IAV strains is underpinned by a diverse TCR repertoire and surprisingly that no shared clonotypes could be detected between individuals. This highlights that cross-reactivity and TCRαβ repertoire diversity may provide further protection against distinct IAV strains. Furthermore, TCRαβ diversity is likely to aid in limiting viral escape, offering protection against any quasispecies that arise during infection. Additionally, we have shown that cross-reactivity towards distinct IAV strains is not limited to HLA-B37-$NP_{338}$, and cross-reactive TCRs towards the HLA-A1-restricted $NP_{44}$ were also observed. In both contexts, we have isolated cross-reactive clonotypes that were not shared between individuals, showing the plasticity of the TCR repertoire whereby multiple TCRs are able to cross-react and provide protection against multiple IAV strains.

Overall, our study provides evidence that inter-epitope cross-reactivity is common for the human influenza epitopes and occurs for prominent HLA-B37-restricted $NP_{338}$ and HLA-A1-restricted $NP_{44}$ epitopes, in addition to previously published HLA-$A^*$02:01-restricted $M1_{58}$[30]. $CD8^+$ T cells can recognize

variations within $CD8^+$ T cell epitopes and cross-reactive $CD8^+$ TCRαβ repertoires can recognize variant HLA-B37-restricted $NP_{338}$ and HLA-A1-restricted $NP_{44}$ peptides across donors. Together, these data highlight the importance of $CD8^+$ T cell cross-reactivity and diversity in the protection against distinct IAV strains and the induction of such diverse and cross-reactive $CD8^+$ T cells by a $CD8^+$ T cell-mediated vaccine could provide universal and superior protection against distinct IAV strains.

## Methods

**Ethics statement**. All work was undertaken in line with the Australian National Health and Medical Research Council (NHMRC) Code of Practice, with ethics approval by the University of Melbourne Human Ethics Committee, ethics numbers 0931311 and 1443389. Australian Red Cross Blood donors provided written informed consent on the day of their blood donation. Written informed consent was obtained from all healthy blood donors. Donor information including HLA typing result is reported in Supplementary Table 3.

**$NP_{338}$ conservation analysis**. Full-length vaccine, Australian, pH1N1, H5N1, and H7N9 isolates with identical sequences collapsed were obtained from the NCBI Influenza Research Database https://www.ncbi.nlm.nih.gov/genomes/FLU/Database/nph-select.cgi?go=database in February 2018. Sequences were aligned using https://www.fludb.org/brc/home.spg?decorator=influenza.

**Antigen-presenting cell lines (APC lines)**. APC lines were kindly provided by Professor James McCluskey (University of Melbourne, VIC, Australia; C1R-$B^*$44:03) and Dr. Nicole Mifsud (Monash University, VIC, Australia; C1R-$B^*$18:01). The A2-20091204 (HLA-$B^*$37:01$^+$) BLCL cells were provided by Professor Weisan Chen (LaTrobe University, VIC, Australia).

**Isolation of PBMCs**. Buffy coats were obtained from the Australian Red Cross Blood Service (ARCBS) and whole blood donations from healthy volunteers. PBMCs were separated using density gradient separation and were cryopreserved until use. All samples were HLA-typed by the Victorian Transplant and Immunogenetics Service (VTIS, West Melbourne, VIC, Australia) at the ARCBS tissue-typing Laboratory.

**Generation of $NP_{338}$-specific $CD8^+$ T cell lines**. PBMCs were used to generate $CD8^+$ T cell lines at a 1:2 stimulator-to-responder ratio. Stimulator PBMCs were washed in fetal calf serum (FCS)-free media and were pulsed with 10 μM (unless otherwise stated) peptide for 90 min at 37 °C with 5% $CO_2$. Stimulator PBMCs were washed twice and added to the responder PBMCs. Cells were cultured for 10–16 days and supplemented twice weekly with 10 U/mL rIL-2 (Roche, Basel, Switzerland) from day 4.

**Intracellular cytokine staining**. APC lines expressing HLA-$B^*$37:01, HLA-$B^*$18:01 or HLA-$B^*$44:03 were pulsed with 10 μM peptide (unless otherwise stated) in FCS-free media for 1 h at 37 °C with 5% $CO_2$. APCs were subsequently washed twice and added to $CD8^+$ T cell lines at 1:2 APC-to-T cell ratio. Alternatively, $CD8^+$ T cell lines from HLA-$B^*$44:02$^+$ donors were stimulated directly with 1 μM peptide without APCs. Cells were co-cultured for 5–6 h in the presence of 10 U/mL IL-2, 5 μM Monensin A (Sigma Aldrich, MO, USA) and Golgi-Stop (BD eBiosciences, CA, USA) with αCD107a-FITC/AF488 (1:100–1:200; eBiosciences # 53-1079-41, CA, USA) for 5 h. Following activation, cells were surface stained for 30 min with αCD3-PE (1:25–1:50; BD Pharmingen # 555333, CA, USA), αCD8-APC (1:50–1:100; BD Biosciences # 340584, CA, USA) with or without Live/Dead-NIR (1:1000; Molecular Probes Cat#L10119, MA, USA), or αCD3-Pey7 (1:50–1:100; eBiosciences # 25-0038-42), αCD8-PerCPCy5.5 (1:50; BD Biosciences # 341051) and Live/Dead-NIR (1:1000; Molecular Probes #L10119). Cells were then fixed with 1% paraformaldehyde (PFA; Electron Microscopy Sciences, PA, USA) or BD-Fix-Perm buffer (BD Biosciences) for 15–20 min. Cells were intracellularly stained for 30 min with anti-IFNγ-PerCPCy5.5 (1:100; eBiosciences #45-7319-41) and anti-TNFα-PeCy7 (1:100; BD Pharmingen #557647, CA, USA), anti-IFNγ-PE (1:40; BD Biosciences #340452) and anti-TNFα-APC (1:50–1:100; BD Biosciences #340534) or anti-IFNγ-V500 (1:50–1:100; BD Horizon #561980, NY, USA) and anti-TNFα-APC (1:50–1:100; BD Biosciences #340534) in perm wash buffer (BD Biosciences), or 0.3% Saponin (Sigma, MO, USA), respectively. Cells were washed, acquired on the BD FACS Canto II (BD Biosciences) and analyzed using the Flowjo software version 9.7.6 (Treestar, OR, USA).

**Tetramer staining of $CD8^+$ T cell lines**. $CD8^+$ T cells were stained with tetramers for 1 h at room temperature in the dark. Cells were washed and surface stained for 30 min with αCD3-PeCy7 (1:50–1:100; eBiosciences #25-0038-42) or αCD3-PB (1:50–1:100; Biolegend #300431, SD, USA) with αCD8-PerCPCy5.5 (1:50; BD Biosciences #341051), αCD27-APC (1:50–1:100; BD Biosciences #337169), αCD45RA-FITC (1:50–1:100; BD Pharmingen #555488) and Live/Dead-

NIR (1:1000; Molecular Probes #L10119) at 4 °C, then either fixed with 1% PFA (Electron Microscopy Sciences), and acquired on the BD FACS Canto II (BD Biosciences) or resuspended in sort buffer and single-cell sorted on the BD Aria III (BD Biosciences).

**Ex vivo magnetic enrichment of PBMCs.** In all, $1–2 \times 10^7$ PBMCs from HLA-B*37:01+ donors were FcR blocked in MACS buffer (phosphate-buffered saline (PBS), 0.5% bovine serum albumin (BSA); Gibco, CA, USA, 0.2 nM EDTA; Ajax Finechem, NSW, Australia) for 15 min at 4 °C (Miltenyi Biotech, Bergisch Gladbach, Germany) and tetramer stained with variant-specific tetramers conjugated to PE in MACS buffer for 1 h at room temperature. Cells were washed, a small amount was removed for unenriched control, and labeled with anti-PE microbeads (Miltenyi Biotech, Bergisch Gladbach, Germany) for 30 min at 4 °C. Following washing, cells were enriched by passing twice over LS magnetic columns (Miltenyi Biotech, Bergisch Gladbach, Germany)[30,55]. Bound cells were eluted in MACS buffer. All samples (unenriched, enriched, and flow-through) were surface stained for 30 min with αCD3-PeCy7 (eBiosciences), αCD8-PerCPCy5.5 (BD Biosciences), Live/Dead-NIR (Molecular Probes), αCD14-APCH7 (BD Pharmingen), αCD4-APCH7 (BD Biosciences), αCD19-APCH7 (Biolegend), αCD27-APC (BD Biosciences) and αCD45RA-FITC (BD Pharmingen) at 4 °C in the dark. Lymphocytes were washed, re-suspended in sort buffer (PBS, 0.1% BSA; Gibco, CA, USA), and were acquired on the BD Aria III.

**Single-cell multiplex RT-PCR for TCRαβ analysis.** Epitope-specific CD8+ T cells from PBMCs or CD8+ T cell lines were tetramer and antibody stained, as described above. Following surface staining, cells were re-suspended in sort buffer (PBS +0.1% BSA). Single epitope-specific CD8+ T cells were sorted on the BD FACS Aria III (BD Biosciences) directly into PCR plates (Eppendorf, Hamburg, Germany). cDNA was synthesized using the VILO RT Kit at 1/20 of the manufacturer's recommendation, supplemented with 0.1% Triton X (Sigma, MO, USA)[30,37,56]. The resulting cDNA was subject to a nested PCR containing 40-Vα and 27-Vβ primers and Cα and Cβ primers (Supplementary Table 9). PCR products were purified using Exo-Sap or Exo-Star (GE Healthcare, Buckinghamshire, UK) and were sequenced using BigDyeV3.1 (Applied Biosystems, CA, USA). Sequencing products were cleaned using DyeEx sequencing plates (Qiagen, Limburg, Netherlands) and sequencing was performed by the Department of Pathology at the University of Melbourne (Melbourne, VIC, Australia). Sequences were analyzed using Finch TV (Geospiza, WA, USA) and the IMGT query software[57,58] (www.imgt.org/IMGT_vquest). CDR3 amino acid sequences described within the text are productive (no stop codons and an in-frame junction) TCRαβ pairs and start from CDR3-position 1, as determined by the IMGT query software. CDR3 length is calculated from position 4 and excludes the final xF motifs. TRAV and TRBV nomenclature was derived from IMGT www.imgt.org/IMGTrepertoire/LocusGenes/#J.

NP338+CD8+ T cells with two alpha or beta chains were dissected by performing the internal round of PCR with the specific individual primers and subsequently sequenced using the relevant constant reverse primer or variable-specific primers (Supplementary Table 9). Where possible, both chains were resolved; however, in a few instances only the dominant chain could be resolved. TRBV19+ TCRβs were sequenced with the TRBV19 forward primer (and are thus referred to as TRBV19) and CDR3 regions were determined using the EXPASY translate tool.

In cases when NP338+CD8+ TCRα or TCRβ chains could not be resolved, sequences (two cells per clonotype) were resolved by cloning and colony PCR. PCR products were ligated into a pGEM-T Easy Vector (Promega, WI, USA) overnight at 16 °C. DH5α cells were then transformed with this vector and were grown, plated on Luria agar plates (containing 100 μg/mL ampicillin, Media Preparation Unit, Department of Microbiology and Immunology, University of Melbourne, VIC, Australia) for 16–18 h, and were selected using X-gal (800 μg). Individual transformed colonies were amplified by PCR with 20 pmol sense and anti-sense vector primers (Supplementary Table 9). PCR were amplified as follows: 1 cycle of 95 °C for 5 min, 30 cycles of 95 °C for 30 s, 57 °C for 30 s, and 72 °C for 1 min, followed by 1 cycle of 95 °C for 1 min, 57 °C for 1 min, and 72 °C for 7 min. Products were visualized on a 2% agarose gel and were sequenced as above with 5 pmol of sense vector primer (Supplementary Table 9).

**Protein expression, purification, and crystallization.** The EM2 TCR contained an engineered disulfide linkage in the constant domains between the TRAC and TRBC. The α- and β-chains were expressed separately as inclusion bodies in a BL21 *Escherichia coli* strain. The inclusion bodies were washed and suspended in 6 M guanidine, then mixed into a cold refolding solution containing 5 M urea, 100 mM Tris-HCL (pH 8), 2 mM EDTA, 400 mM L-arginine-HCl, 0.5 mM oxidized glutathione, and 5 mM reduced glutathione. The refolding solution was dialyzed after 3 days against 10 mM Tris-HCl (pH 8) and 150 mM NaCl and then purified using anion exchange and size exclusion columns[59]. All peptides were purchased from Genescript (NJ, USA). HLA-B*37:01+-restricted peptides include WT-NP338-346 (FEDLRVLSF), L7S-NP338-346 (FEDLRVSSF), and V6L-NP338-346 (FEDLRLLSF). Soluble class I heterodimers (with or without BirA tag at the C-terminal) of HLA-B*18:01 and HLA-B*44:05 containing the NP338 peptides and HLA-B*37:01

containing the NP338 and variant NP338-L7S and NP338-V6L peptides were prepared similarly to the TCR without urea in the refolding solution[59]. Crystals of the EM2 TCR–HLA-B*37:01-NP338 complex or the pHLA individually in 10 mM Tris-HCl (pH 8.0) and 150 mM NaCl were grown by the hanging-drop, vapor-diffusion method at 20 °C with a protein to reservoir drop ratio of 1:1 and a protein concentration of 6–10 mg/mL. The pHLA crystals formed in 20% PEG 6000, 0.2 M NaCl, 0.1 M Na citrate pH 6.5 while the EM2 TCR-HLA-B*37:01-NP338 complex crystals formed in 19% PEG 3350 and 0.2 M di-ammonium tartrate. Monomers of pHLA with the BirA tag construct were biotinylated and tetramerized in a 1:4 ratio with PE-streptavidin (Life Technologies, CA, USA) in 6 additions over the course of 1 h.

**Data collection and structure determination.** Crystals were soaked in a cryo-protectant solution containing mother liquor solution with the PEG concentration increased to 30% (weight per volume) and then flash frozen in liquid nitrogen. Data were collected on the MX1 beamline at the Australian Synchrotron, Clayton using the ADSC-Quantum 210 CCD detector (at 100 K)[60]. Data were processed using the XDS software[61] and scaled using the XSCALE software[61]. The pHLA and EM2 TCR-HLA-B*37:01-NP338 complex structures were determined by molecular replacement using the PHASER[62] program with the HLA-A*02:01-NLV complex as the search model for the HLA without the peptide (Protein Data Bank accession number, 3GSO[59]) and the LC13 TCR for the TCR (Protein Data Bank accession number, 1KGC[63]). Manual model building was conducted using the Coot software[64] followed by maximum-likelihood refinement with PHENIX[65] and Buster programs[66]. The TCR was numbered using the IMGT nomenclature[57]. The final models have been validated using the Protein Data Base validation website and the final refinement statistics are summarized in Supplementary table 2. Coordinates were validated by the PDB database. All molecular graphic representations were created using PyMol[67].

**Thermal stability assay.** The thermal stability assay was performed in the Real Time Detection system (Corbett RotorGene 3000) using the fluorescent dye Sypro orange to monitor protein unfolding. The pHLA complexes (5 and 10 μM) in 10 mM Tris-HCl pH 8 and 150 mM NaCl were heated from 30 to 95 °C with a heating rate of 1 °C/min. The fluorescence intensity was measured with excitation at 530 nm and emission at 555 nm. The thermal melt point (Tm) represents the temperature for which 50% of the protein is unfolded.

**Surface plasmon resonance measurement and analysis.** Surface plasmon resonance experiments were conducted at 25 °C on the BIAcore 3000 instrument with TBS buffer (10 mM Tris-HCl, pH 8, 150 mM NaCl, and 0.005% surfactant P20). TBS buffer was supplemented with 1% BSA to prevent non-specific binding. The human TCR-specific monoclonal antibody, 12H8 antibody[68], was coupled to research-grade CM5 chips with standard amine coupling. The experiment was conducted with one empty flow cell (use for subtraction as blank) and one flow cell with the EM2 TCR captured to approximately 300 response unit per flow cell onto the 12H8 antibody[68]. Then a concentration range of 200 μM maximum of the pHLA complexes were passed over all the flow cells. The final response was calculated by subtracting the response of the antibody alone and that of EM2 TCR. BIAevaluation Version 3.1 was used for data analysis using the 1:1 Langmuir binding model. Experiments were carried out in duplicate ($n = 2$).

**Statistical analyses.** All statistical analyses were undertaken using Prism 6 (GraphPad, CA, USA). A Dunnett's two-way analysis of variance was used for all non-parametric non-paired analysis where multiple comparisons were made. Statistical significance was defined as $*p \leq 0.05$, $**p \leq 0.01$, and $***p \leq 0.001$.

## Data availability

Coordinates submitted to PDB database, and the PDB codes are 6MT6 (HLA-B*37:01-NP338), 6MT4 (HLA-B*37:01-NP338-L7S), 6MT5 (HLA-B*37:01-NP338-V6L), 6MTL (HLA-B*44:05-NP338), 6MT3 (HLA-B*18:01-NP338), and 6MTM (EM2 TCR-HLA-B*37:01-NP338). All other data that support the findings of this study are available from the corresponding author upon reasonable request. A reporting summary for this article is available as a Supplementary Information file.

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

## Acknowledgements

We thank Hanim Halim, the Monash Macromolecular Crystallization Facility staff and the staff at the Australian synchrotron for technical assistance. This work was supported by Australian National Health and Medical Research Council (NHMRC) Project (AI1008854) and Program (AI1071916) Grants awarded to K.K. E.J.G. is supported by an Early Career NHMRC CJ Martin Fellowship; J.R. by an Australian Research Council (ARC) Laureate fellowship; S.G. is a Monash Senior Research Fellow; and K.K. is supported by an NHMRC SRF Level B Fellowship.

## Author contributions

Conceived and designed the experiments: E.J.G., K.K., S.G., M.B., W.C. Performed the experiments: E.J.G., E.B.C., L.L., S.G. Analyzed the data: E.J.G., T.J., L.L., S.S., E.B.C., S.G. Wrote the paper: E.J.G., T.J., M.B., J.R., W.C., K.K., S.G.

## Additional information

**Competing interests:** The authors declare no competing interests.

