## [Peer Review File · Nature Communications]

REVIEWERS' COMMENTS:

Reviewer #1 (Remarks to the Author):

In this manuscript, Grant et al. identify and characterize TCR clonotypes able to recognize the NP338 peptide of influenza A virus presented by HLA-B37. They show that some of these clonotypes can also recognize certain natural variants of the NP338 peptide, in particular NP338-L7S and NP338-V6L. This is not especially surprising since NP338-L7S and NP338-V6L each differ from NP338 by a one relatively conservative amino acid substitution. Indeed, superposition of the NP338/HLA-B37, NP338-L7S/HLA-B37 and NP338-V6L/HLA-B37 crystal structures show that the peptide conformations are virtually identical, except at the substituted position. The structure of a cross-reactive TCR (EM2) bound to NP338/HLA-B37 provides a straightforward explanation for its ability to accommodate these peptide variants.

It was the term "molecular mimicry" in the title that initially piqued my curiosity. However, use of this term to describe a simple case of cross-reactivity in which wild-type and variant peptides differ by a single residue is not justified. The cellular and structural studies described here are technically very solid. However, there is nothing particularly novel or remarkable in the authors' findings. Therefore, this manuscript is not suitable for *Nat. Commun.* It is more appropriate for a specialized journal such as *J. Immunol.*

Reviewer #2 (Remarks to the Author):

This is an important well written manuscript which elegantly links the structural features of variant influenza A CD8 T cell epitopes and selection of TCR repertoires that are crossreactive in less well studied HLA-B37 and A1 responses based on molecular mimicry. Importantly, these results suggest that CD8 T cell target universal influenza A vaccines could be generated and provide protection against different strains. There were only a few minor issues that would need to be addressed:

1 Introduction, page 3: no references were added in the part about analysis of M1-specific TCR repertoires.

2 Results (page 6): typo "NP366" instead of "NP338".

3 Results (page 7): in description of Fig 2 authors do not mention that the tetramer staining and functional analysis actually works best for V6L variant. This point is important to emphasize in the beginning since authors explain it later on page 11. There is no information in the results section about cultures stim wt peptide and re-stim with it, for both tet staining and ICS.

Also if the scale in Fig 2c is adjusted it would be easier to see the differences within all groups.

4 Fig 4b is unclear the Y-axis is that (% of CD8 T cells) as these numbers are much lower than what appears in the FACS plots in Fig 4a.

4 Supplementary Table7: D21- frequency of sequences from wt peptide only, not S7N variant.

Discussion (page 14): the paragraph about diversity and cross-reactivity is confusing especially in the sentence starting with "Our data highlight...". Based on their own studies they showed that both diversity and crossreactivity are important and then suggest that if crossreactivity is absent the diversity provides protection. This paragraph should emphasize both features equally since their data proved that clearly. To help emphasize that T cell crossreactivity can play an important role in protective immunity even between different pathogens it would be helpful to reference manuscripts such as Watkin LB et al *JACI* 2017, which shows that IAV-specific cells with a particular TCR may actually protect against infection with an unrelated virus like EBV or reviews on heterologous immunity such as Welsh RM, Che JW, Brehm MA, Selin LK., *Immunol Rev.* 2010.

Reviewer #3 (Remarks to the Author):

This is a detailed and elegant study of the CD8 T cell response to influenza NP epitopes; the study focuses mostly on one specific epitope presented by HLA *37:01, and also considers more briefly an additional A1*01 restricted NP epitope.

The study combines functional, sequence and structural data. The study provides one of the most detailed pictures of how a set of TCRs may recognise a peptide and its variants. It highlights the enormous diversity and plasticity of the T cell immune repertoire. The study is clearly written and presented, and the data seem sound and to support the major conclusions of the study.

I may have missed this, but I could not find any details of the deposited structure PDB IDs, or any other way to get the relevant co-ordinate data sets.

RESPONSES TO REVIEWERS' COMMENTS

We thank the Reviewers for their constructive comments and appreciating the importance of our study.

We responded to Reviewers' questions and comments in a point-by-point form, and we have amended the manuscript and highlighted the changes in yellow.

Reviewer #1 (Remarks to the Author):

It was the term “molecular mimicry” in the title that initially piqued my curiosity. However, use of this term to describe a simple case of cross-reactivity in which wild-type and variant peptides differ by a single residue is not justified.

Following the Reviewer's comment, we have modified the manuscript accordingly and removed the term “molecular mimicry” throughout the manuscript.

Reviewer #2 (Remarks to the Author):

We thanks the Reviewer for the positive comments and feedback. We have addressed the minor comments, as outlined below.

1 Introduction, page 3: no references were added in the part about analysis of M1-specific TCR repertoires.

This has been added on page 3, reference 30.

2 Results (page 6): typo “NP366” instead of “NP338”.

We have replaced “NP366” with “NP338” on page 6.

3 Results (page 7): in description of Fig 2 authors do not mention that the tetramer staining and functional analysis actually works best for V6L variant. This point is important to emphasize in the beginning since authors explain it later on page 11.

We agree with the Reviewer that the V6L tetramer appears to stain better than the WT tetramer, however, there is no difference between the WT and V6L tetramer staining.

Following the Reviewer's comment, we added a statement to flag this observation on page 7, as we agree this is important for the analysis of results on page 11 and the SPR data. We have amended the manuscript accordingly (page 7):

Interestingly, the NP₃₃₈-V6L tetramer showed a slightly stronger staining than the WT and NP₃₃₈-L7S tetramers (**Fig. 2a**).

There is no information in the results section about cultures stim wt peptide and re-stim with it, for both tet staining and ICS. Also if the scale in Fig 2c is adjusted it would be easier to see the differences within all groups.

The information related to the peptide culture methodology is provided in Figure Legend and Methods sections (page 15; CD8⁺ T cell lines, page 16; ICS). Furthermore, as the range of CD8⁺ T cell responses between the donors varies greatly (~60% Donor 1 and ~4% Donor 3), we feel the individual scales in Fig. 2c most accurately represent the level of cross-reactivity towards the variant peptides across individual donors.

4 Fig 4b is unclear the Y-axis is that (% of CD8 T cells) as these numbers are much lower than what appears in the FACS plots in Fig 4a.

Figure 4b represents the frequency of epitope-specific CD8⁺ T cells within the total CD8⁺ T cell population, and not the % of cells. We clarified this point in the text and amended accordingly (page 8):

Interestingly, NP₃₃₈-V6L-specific CD8⁺ T cells were found at the highest average frequency, at 1 in every 5.97×10^{-4} CD8⁺ T cells, followed by NP₃₃₈ and then NP₃₃₈-L7S-specific CD8⁺ T cells found at frequencies of 1 in 2.31×10^{-4} and 1.86×10^{-4} CD8⁺ T cells, respectively (Fig.4b).

4 Supplementary Table7: D21- frequency of sequences from wt peptide only, not S7N variant.

Donor 21 did not have any detectable S7N⁺CD8⁺ T cell population, as indicated in Fig. 8b (previously Fig8c). To further clarify this point, we have modified the following sentence in Results (page 12):

Two of three donors (Donors 1 and 20) had both NP₄₄- and NP₄₄-S7N-specific CD8⁺ T cell populations. Interestingly, Donor 21 had only a NP₄₄-specific CD8⁺ T cell population, with no detectable S7N-specific CD8⁺ T cell population (Fig. 8b).

Discussion (page 14): the paragraph about diversity and cross-reactivity is confusing especially in the sentence starting with “Our data highlight...”. Based on their own studies they showed that both diversity and crossreactivity are important and then suggest that if crossreactivity is absent the diversity provides protection. This paragraph should emphasize both features equally since their data proved that clearly.

We thank the Reviewer for their input and the opportunity to further emphasise this section in Discussion. We have modified page 14 accordingly:

Our data show that cross-recognition of distinct IAV strains is underpinned by a diverse TCR repertoire, and surprisingly that no shared clonotypes could be detected between individuals. This highlights that cross-reactivity and TCR repertoire diversity may provide further protection against distinct IAV strains.

Reviewer #3 (Remarks to the Author):

We thank the Reviewer for their positive review of our manuscript. We now added the PDB codes to the manuscript for our structures in the “Data Availability” section. We have updated page 21 accordingly:

Coordinates submitted to PDB database, and the PDB codes are 6MT6 (HLA-B*37:01-NP₃₃₈), 6MT4 (HLA-B*37:01-NP₃₃₈-L7S), 6MT5 (HLA-B*37:01-NP₃₃₈-V6L), 6MTL (HLA-B*44:05-NP₃₃₈), 6MT3 (HLA-B*18:01-NP₃₃₈), and 6MTM (EM2 TCR-HLA-B*37:01-NP₃₃₈).